# Beyond Re-Training from Scratch: Exploiting the Pre-Trained Classifier for Long-Tailed Learning

## Abstract

Fine-tuning for long-tailed learning has garnered significant interest owing to the strong priors in foundation models. A prevailing approach is to explore various long-tailed strategies under the standard fine-tuning paradigm, in which the model is initialized from the pre-trained backbone, while the pre-trained classifier is discarded and replaced with a newly re-trained one. However, we observe that, under tail data scarcity, this newly re-trained classifier suffers from weakened discriminative ability and semantic awareness, exhibiting severe imbalance in class-discriminative channels and mislearning general features for tail classes; in contrast, the pre-trained classifier behaves much closer to the oracle, highlighting its strong potential as an effective guide. Motivated by this, we propose a new fine-tuning paradigm, `PTClf` (Pre-Trained Classifier helps), which exploits the pre-trained classifier to assist the re-trained one in learning tail classes. Specifically, we first align downstream and upstream classes through label mapping, and then guide the re-trained classifier to learn from the mapped pre-trained classifier via initialization and regularization, thus facilitating knowledge transfer from related upstream classes to the data-scarce tail classes. Experiments show that `PTClf` delivers remarkable benefits for long-tailed data, especially for tail classes, while also exhibiting strong versatility in few-shot learning and domain generalization.

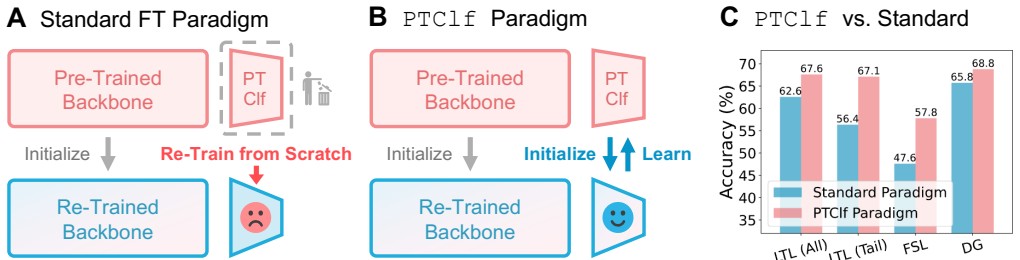

Figure 1: **The pre-trained classifier can significantly benefit downstream tasks.** **A**: The default fine-tuning paradigm transfers upstream knowledge merely from the pre-trained backbone, while discarding the pre-trained classifier and re-training a new one from scratch. **B**: Our proposed paradigm, `PTClf`, further leverages knowledge within the pre-trained classifier to assist the re-trained classifier. **C**: With the help of the pre-trained classifier, `PTClf` delivers notable improvements on long-tailed data, as well as in few-shot learning (FSL) and domain generalization (DG).

## 1 Introduction

Real-world classification tasks often exhibit a long-tailed distribution over classes, where a few head classes contain abundant samples while most tail classes contain only a limited number (Van Horn et al., 2018; Zhang et al., 2023). Owing to this scarcity of samples, naïve learning struggles to generalize on tail classes and tends to be heavily biased toward head classes (Menon et al., 2021). Motivated by this, long-tailed learning (LTL) has been widely studied to mitigate such challenges and achieve more balanced performance (Cao et al., 2019; Wang et al., 2021; Hou et al., 2025).

Recently, foundation models, *i.e.*, large-scale pre-trained neural networks, have proven highly effective at transferring upstream knowledge to alleviate the data scarcity issue in tail classes (Tian et al., 2022; Dong et al., 2023; Zhu et al., 2023a; Zhao et al., 2024b; Shi et al., 2024b). The most straightforward approach to such knowledge transfer is to directly fine-tune on these foundation

models (Yosinski et al., 2014; Yu et al., 2024). Therefore, a series of methods has explored various long-tailed training strategies under the standard fine-tuning paradigm, such as two-stage learning (Ma et al., 2021; Dong et al., 2023) and ensemble learning (Ru et al., 2024), aiming to leverage upstream knowledge without incurring bias. Despite showcasing strong performance, these existing approaches rely heavily on the standard fine-tuning paradigm, transferring knowledge from the pre-trained backbone while overlooking the pre-trained classifier, as illustrated in Fig. 1.

Discarding the pre-trained classifier and re-training a new one from scratch is the default practice in fine-tuning; however, this paradigm implicitly assumes that the downstream task contains abundant and balanced data to train a reliable classifier. LTL obviously does not satisfy this assumption. Numerous prior studies have emphasized that the classifier itself is often the main culprit in LTL due to the scarcity of tail-class data (Kang et al., 2020; Zhang et al., 2021; Yang et al., 2022; Alshammari et al., 2022). In this light, the neglected pre-trained classifier, embedded with rich upstream knowledge, may instead be an antidote. Therefore, two interesting questions arise:

> *What happened to the re-trained classifier in long-tailed learning with foundation models?*

> *Can we leverage the pre-trained classifier to help the re-trained classifier?*

Our work is motivated by the above questions to revisit the classifiers in the context of LTL with foundation models. We begin by investigating the channel behavior within the re-trained and pre-trained classifiers (refer to § 3). We observe that, class-discriminative channels (*i.e.*, the classifier factors crucial for distinguishing particular classes) are highly sparse in tail classes but consistently dense in head classes, explicitly revealing that **the re-trained classifier's discriminative ability is less effective for tail classes** (*cf.* Fig. 3). Moreover, we further observe that, the class-discriminative channels of tail classes rely excessively on general feature channels (*i.e.*, the features broadly activated for all classes), indicating that **the re-trained classifier's semantic awareness is weakened and results in mislearning general factors for tail classes instead of class-specific ones** (*cf.* Fig. 4). Notably, we find that data scarcity of tail data is the root cause of both issues, whereas **the pre-trained classifier, trained on large-scale data, exhibits behavior closely aligned with the oracle and demonstrates strong potential to mitigate them**.

Inspired by this, we propose a new fine-tuning paradigm, termed `PTClf` (Pre-Trained Classifier helps), which exploits the strengths of the pre-trained classifier to help the re-trained classifier in learning tail classes. Specifically, the proposed `PTClf` method first aligns each downstream class with its most similar upstream counterpart via label mapping. Using this mapping, `PTClf` initializes the re-trained classifier with the mapped pre-trained one, and further regularizes it to continue learning from the pre-trained classifier during fine-tuning. As such, `PTClf` enables the pre-trained classifier to effectively transfer knowledge from related upstream classes to the re-trained classifier, thereby benefiting the data-scarce classes.

We summarize our main observations and contributions as follows: **(1)** We observe that the re-trained classifier exhibits severe imbalance in class-discriminative channels and further mislearns general features as discriminative for tail classes, demonstrating weakened discriminative ability and semantic awareness. **(2)** In contrast, we observe that the pre-trained classifier behaves much closer to the oracle, underscoring its strong potential as an effective guide. **(3)** Motivated by this, we propose a new fine-tuning paradigm, termed `PTClf`, which leverages upstream knowledge within the pre-trained classifier to assist the re-trained one in learning tail classes. **(4)** Extensive experiments demonstrate that `PTClf` consistently enhances performance across various downstream tasks (*cf.* Fig. 1), particularly on the long-tailed benchmark, attaining an average gain of 5.0 pp in overall accuracy and up to 10.8 pp on tail classes over the baseline.

## 2 RELATED WORK

**Classifier Dilemma in LTL.** Numerous studies have highlighted that the classifier is often the main culprit in LTL (Kang et al., 2020; Zhang et al., 2019; Menon et al., 2021; Sun et al., 2025). A representative study by Kang et al. (2020) first revealed that the backbone learned by naïve training is already high-quality, but the classifier itself is heavily biased and constitutes the main cause of performance degradation. Specifically, they observed that the classifier's weight norms of head classes are significantly larger than those of tail classes, resulting in skewed logits with erroneously

higher confidence scores for head classes. To address this, a line of research has developed various approaches to balance classifiers' norms, *e.g.*, directly normalizing the norms (Kang et al., 2020) or penalizing excessively large ones (Alshammari et al., 2022). However, existing methods remain confined to a norm-level perspective of the classifier. This paper, in contrast, introduces a channel-level analysis, offering a deeper investigation into the classifier's behavior.

**LTL with Foundation Models.** Foundation models have recently demonstrated their effectiveness in LTL by facilitating knowledge transfer and improving long-tailed performance (Dong et al., 2023; Shi et al., 2024a). Dong et al. (2023), specifically, show that upstream knowledge can be effectively aligned to downstream tasks via fine-tuning, allowing tail classes to benefit and achieve stronger results. Therefore, a line of research has explored various long-tailed strategies within the standard fine-tuning paradigm, *e.g.*, decoupling naïve and balanced training stages (Ma et al., 2021; Dong et al., 2023) or introducing specialized experts for head and tail classes (Ru et al., 2024). However, existing methods solely transfer knowledge to the backbone while overlooking the classifier, which is the main devil in LTL. This paper, in contrast, leverages the knowledge encoded in the pre-trained classifier to guide the re-trained classifier, thereby achieving superior long-tailed performance.

**Parameter-Efficient Fine-Tuning.** This paper mainly focuses on parameter-efficient fine-tuning (PEFT), since Shi et al. (2024a) show that full fine-tuning can significantly hurt tail classes, whereas lightweight fine-tuning better preserves upstream knowledge and benefits them. PEFT approaches (Xin et al., 2024) can be categorized as: reparameter tuning (Hu et al., 2022; Jie & Deng, 2023), which integrates newly learnable parameters to the pre-trained weights; prompt tuning (Jia et al., 2022; Tang et al., 2025), which injects learnable tokens into vision Transformer (ViT) blocks; adapter tuning (Steitz & Roth, 2024; Chen et al., 2022), which inserts separate learnable modules within ViT blocks. The proposed `PTClf` is compatible with various fine-tuning approaches.

## 3 A CLOSER LOOK AT CLASSIFIERS IN LONG-TAILED LEARNING

In this section, we first introduce two channel-wise metrics, and then systematically analyze the channel behavior of the re-trained and pre-trained classifiers in LTL.

**Notations.** We define the notations used in this paper. Let $\mathcal{T} = \{(\mathbf{x}_i, y_i^{\mathrm{r}})\}_{i=1}^N$ be the downstream training set, where sample $\mathbf{x}_i$ is labeled as $y_i^{\mathrm{r}} \in [K^{\mathrm{r}}]$. For class $k^{\mathrm{r}}$, $\mathcal{T}_{k^{\mathrm{r}}} \subset \mathcal{T}$ is the set of its training samples and $|\mathcal{T}_{k^{\mathrm{r}}}|$ is its cardinality. Without loss of generality, we assume $|\mathcal{T}_1| \geq |\mathcal{T}_2| \geq \ldots \geq |\mathcal{T}_{K^{\mathrm{r}}}|$. The imbalance ratio is defined as $\mathrm{IR} = |\mathcal{T}_1|/|\mathcal{T}_{K^{\mathrm{r}}}|$. Let $f_{\boldsymbol{\theta}}(\cdot)$ be the fine-tuned ViT backbone parameterized by $\boldsymbol{\theta}$. We denote $\mathbf{z}_i = f_{\boldsymbol{\theta}}(\mathbf{x}_i) \in \mathbb{R}^D$ as the feature and $\mathbf{Z} = [\mathbf{z}_1, \mathbf{z}_2, \ldots, \mathbf{z}_N] \in \mathbb{R}^{N \times D}$ as the feature matrix. The final class prediction is given by a linear *re-trained* classifier $\hat{y}_i^{\mathrm{r}} = \arg\max(\mathbf{W}^{\mathrm{r}}\mathbf{z}_i + \mathbf{b}^{\mathrm{r}})$, where $\mathbf{W}^{\mathrm{r}} = [\mathbf{w}_1^{\mathrm{r}}, \mathbf{w}_2^{\mathrm{r}}, \ldots, \mathbf{w}_{K^{\mathrm{r}}}^{\mathrm{r}}] \in \mathbb{R}^{K^{\mathrm{r}} \times D}$ and $\mathbf{b}^{\mathrm{r}} \in \mathbb{R}^{K^{\mathrm{r}}}$ denote the weight matrix and bias vector. For the *pre-trained* classifier, we denote the corresponding weights and biases as $\mathbf{W}^{\mathrm{p}} \in \mathbb{R}^{K^{\mathrm{p}} \times D}$ and $\mathbf{b}^{\mathrm{p}} \in \mathbb{R}^{K^{\mathrm{p}}}$, where $K^{\mathrm{p}}$ is the number of upstream classes. We use the superscripts $\mathrm{r}$ and $\mathrm{p}$ to distinguish notations associated with re-trained and pre-trained models.

### 3.1 METRIC DESIGN

We examine the classifiers from two perspectives: **discriminative ability** (*i.e.*, how effectively the classifiers discriminate between classes) and **semantic awareness** (*i.e.*, which feature semantics the classifiers rely on). To assess these aspects, we introduce two channel-wise metrics[1]:

**Class-Discriminative Importance (CDI).** CDI aims to identify classifier channels that are important for distinguishing a given class. We define the CDI score of channel $d$ for class $k^{\mathrm{r}}$ as:

$$\mathrm{CDI}_{k^{\mathrm{r}}, d} = \left| w_{k^{\mathrm{r}}, d}^{\mathrm{r}} \right|. \tag{1}$$

Specifically, CDI follows an intuition: in the linear classifier $(\mathbf{w}_{k^{\mathrm{r}}}^{\mathrm{r}}{}^{\top} \mathbf{z}_i + b_{k^{\mathrm{r}}}^{\mathrm{r}})$, weights $w_{k^{\mathrm{r}}, d}^{\mathrm{r}}$ with larger magnitudes often indicate greater discriminative importance for class $k^{\mathrm{r}}$ (Chan & Veas, 2024; Ye et al., 2018). Therefore, a high $\mathrm{CDI}_{k^{\mathrm{r}}, d}$ score indicates that channel $d$ is a strong discriminative factor for class $k^{\mathrm{r}}$ within the classifier.

**General Importance (GI).** GI aims to identify feature channels that are salient for all classes. We define the GI of channel $d$ as:

$$\mathrm{GI}_d = \frac{1}{K^{\mathrm{r}}} \sum_{k^{\mathrm{r}} \in [K^{\mathrm{r}}]} \frac{1}{|\mathcal{T}_{k^{\mathrm{r}}}|} \left\| \mathbf{Z}_{\mathcal{T}_{k^{\mathrm{r}}}, d} \right\|_1. \tag{2}$$

---

[1]In ViT, each dimension $d \in [D]$ corresponds to a channel.

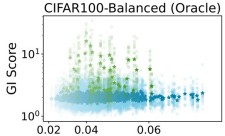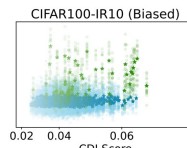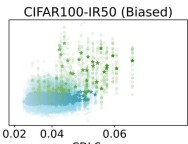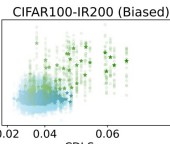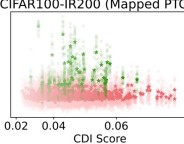

**Figure 3: Behavior of classifiers' class-discriminative channels.** The CDI scores associated with the re-trained and pre-trained classifiers are shown in blue and pink respectively, with darker shades indicating higher CDI. In the biased re-trained classifier, the class-discriminative (high-CDI) channels for tail classes are markedly sparse.

**Figure 4: Behavior of feature channels corresponding to class-discriminative channels.** The GI scores associated with the re-trained and pre-trained classifiers are shown in blue and pink respectively, with top-30 GI channels specifically highlighted in green. In contrast to the oracle re-trained classifier, the biased classifier's class-discriminative (high-CDI) channels for tail classes rely exclusively on general (high-GI) channels.

Specifically, GI is built on the observations by Zhao et al. (2024a): feature channels $\mathbf{Z}_{\mathcal{T}_{k^r},d} \in \mathbb{R}^{|\mathcal{T}_{k^r}|}$ with high norms tend to be broadly informative across different categories. Therefore, a high $\mathrm{GI}_d$ score indicates that channel $d$ is generally important across different classes in the feature space.

**Roles of CDI and GI.** As illustrated in Fig. 2, CDI effectively identifies channels that are crucial for distinguishing the given classes. Therefore, we use the quantity of class-discriminative channels (*i.e.*, high-CDI channels) as a direct measure of the classifier's **discriminative ability** for particular classes. Moreover, Fig. 2 demonstrates that GI identifies channels not only specific to the given classes but also relevant to other classes within the task, highlighting general channels shared across classes (see Appendix A for details). Therefore, we apply GI to investigate the feature semantics of the class-discriminative channels, specifically, whether they correspond to broadly general feature channels (*i.e.*, high-GI channels) or less-general ones, thereby assessing the classifier's **semantic awareness**. A thorough validation of CDI and GI is presented in Appendix A.

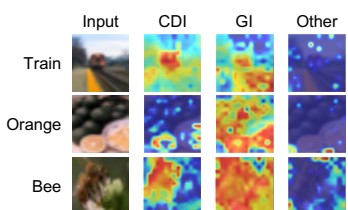

**Figure 2: GradCAM visualization.** High-CDI channels focus on regions specific to particular classes, while high-GI channels tend to capture regions associated with all classes within the task.

### 3.2 MAIN OBSERVATION

We investigate the channel behavior of re-trained and pre-trained classifiers using the above metrics. In particular, we examine the re-trained classifier by comparing the oracle version (trained on the balanced full dataset) and the biased version (trained on an imbalanced subset). Here, we present results on CIFAR100 with CE loss; more results are in Appendix B. Our main observations are:

**Class-Discriminative Channel Imbalance.** In Fig. 3, we visualize the CDI scores of head- and tail-class anchors[2] in the oracle and biased classifiers. For the biased re-trained classifier, it can be clearly observed that the high-CDI (*i.e.*, class-discriminative) channels are significantly sparse in tail classes, with this sparsity increasing as the imbalance ratio grows. By contrast, the high-CDI channels for head classes remain consistently dense under both balanced and long-tailed settings. This disparity explicitly indicates that the discriminative ability in the re-trained classifier is significantly imbalanced, showcasing less effectiveness for tail classes.

> **Observation 1:** The re-trained classifier exhibits severe imbalance in the class-discriminative channels, where those corresponding to tail classes are highly sparse. This leads to the classifier's discriminative ability being far less effective for tail classes than for head classes.

---

[2]For class $k^r$, $\mathbf{w}_{k^r}^r \in \mathbb{R}^D$ is the corresponding class anchor.

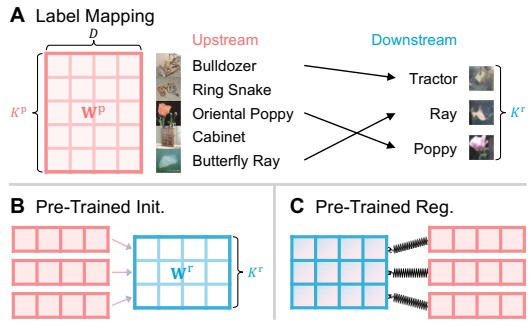

**A** Label Mapping

**B** Pre-Trained Init.

**C** Pre-Trained Reg.

Figure 5: **Overview of PTClf.**

**Algorithm 1** Pseudo-code of PTClf

**Input:** Backbone $f_{\boldsymbol{\theta}}(\cdot)$, pre-trained classifier $\mathbf{W}^{\mathrm{p}}$, long-tailed data $\mathcal{T}$, epochs $E$.
1: Compute the mapping set $\{k^{\mathrm{p}*}(k^{\mathrm{r}})\}_{k^{\mathrm{r}} \in [K^{\mathrm{r}}]}$ via **LM**.
2: Initialize the re-trained classifier $\mathbf{W}^{\mathrm{r}}$ via **PTInit**.
3: **for** $e = 1, 2, \ldots, E$ **do**
4:     Update the mapping set $\{k^{\mathrm{p}*}(k^{\mathrm{r}})\}_{k^{\mathrm{r}} \in [K^{\mathrm{r}}]}$ via **LM**.
5:     Compute the classification loss $\mathcal{L}_{cls}$ by forward propagation.
6:     Compute the regularization loss $\mathcal{L}_{reg}$ via **PTReg**.
7:     Update the model by minimizing $(\mathcal{L}_{cls} + \mathcal{L}_{reg})$.
8: **end for**

**LM**: Label mapping; **PTInit**: Pre-trained initialization; **PTReg**: Pre-trained regularization.

**Class-Discriminative Channel Mislearning.** In Fig. 4, we further visualize the GI scores of the class-discriminative channels for tail-class anchors in the oracle and biased classifiers. We observe that, in the biased re-trained classifier, class-discriminative channels tend to rely heavily on high-GI (general) feature channels as the imbalance ratio increases. This stands in sharp contrast to the oracle classifier, which predominantly leverages the less-general channels as class-specific channels. This explicitly illustrates that the re-trained classifier exhibits weak semantic awareness for tail classes, leading it to mislearn the general channels as the primary class-discriminative factors.

> **Observation 2:** The re-trained classifier tends to mislearn general features as discriminative channels for tail classes, revealing diminished capacity to capture class-specific features and demonstrating weakened semantic awareness.

**Potential Solution from Pre-Training.** The two observations indicate that the naïvely re-trained classifier exhibits imbalanced discriminative ability and suffers from weakened semantic awareness for tail classes. We examine the underlying reasons for these issues in Appendix B. Our analysis reveals that both issues stem from the same root cause: the scarcity of tail-class samples. With limited data, the re-trained classifier fails to acquire sufficient knowledge for adequate discriminative ability and semantic awareness. In contrast, the pre-trained classifier, enriched with abundant upstream knowledge, may instead exhibit more favorable behavior. Interestingly, as shown in Figs. 3 and 4, the pre-trained classifier indeed behaves more closely with the oracle classifier, demonstrating strong potential in alleviating both issues. This naturally leads to a central question: *How can we exploit the pre-trained classifier to help the re-trained classifier?*

## 4 METHODOLOGY

The above analysis demonstrates that the re-trained classifier exhibits undesirable behavior in LTL, while the pre-trained classifier behaves more consistently with the oracle. Here, we introduce a novel paradigm, termed PTClf, leveraging the pre-trained classifier to assist the re-trained classifier. Fig. 5 and Algo. 1 provide an overview and pseudo-code for PTClf.

**Label Mapping.** Since the number of upstream classes $K^{\mathrm{p}}$ typically mismatches that of downstream classes $K^{\mathrm{r}}$ (with $K^{\mathrm{p}} \gg K^{\mathrm{r}}$), it is crucial for a label mapping to align upstream to downstream. We introduce a straightforward one-to-one mapping strategy, which pairs each downstream class with an upstream class based on the prediction frequencies of the pre-trained classifier. Specifically, a downstream class $k^{\mathrm{r}} \in [K^{\mathrm{r}}]$ is mapped to an upstream class $k^{\mathrm{p}} \in [K^{\mathrm{p}}]$ following:

$$k^{\mathrm{p}*}(k^{\mathrm{r}}) = \arg\max_{k^{\mathrm{p}} \in [K^{\mathrm{p}}]} \frac{1}{|\mathcal{T}_{k^{\mathrm{r}}}|} \sum_{\mathbf{x}_i \in \mathcal{T}_{k^{\mathrm{r}}}} \mathbf{1}\{\hat{y}_i^{\mathrm{p}} = k^{\mathrm{p}}\}, \tag{3}$$

where $\hat{y}_i^{\mathrm{p}} = \arg\max(\mathbf{W}^{\mathrm{p}}\mathbf{z}_i + \mathbf{b}^{\mathrm{p}})$ denotes the class prediction given by the pre-trained classifier. To prevent forgetting or bias, the pre-trained classifier is kept frozen during fine-tuning.

**Pre-Trained Initialization.** As demonstrated in **Observation 1**, the class-discriminative channels for tail classes in the re-trained classifier are highly sparse, whereas those for the corresponding classes in the mapped pre-trained classifier exhibit a denser distribution. A natural solution is to leverage the mapped pre-trained classifier for initialization, enabling the re-trained classifier to inherit its well-developed discriminative channels. Concretely, we initialize each class anchor $\mathbf{w}_{k^{\mathrm{r}}}^{\mathrm{r}}$ of the re-trained classifier with the mapped pre-trained class anchor $\mathbf{w}_{k^{\mathrm{p}*}(k^{\mathrm{r}})}^{\mathrm{p}}$.

**Pre-Trained Regularization.** While pre-trained initialization provides the re-trained classifier with strong discriminative ability, it still suffers from weakened semantic awareness noted in **Observation 2** (see Appendix E.1). The issue stems from the suboptimal initial label mapping (*cf.* Fig. 11), which causes the initialized pre-trained anchors to correlate weakly with the downstream classes, leaving a gap in semantics. Nonetheless, as fine-tuning proceeds, the label mapping converges toward an optimal alignment. To enable the re-trained classifier to continually learn from the better-mapped pre-trained classifier and inherit stronger semantic awareness ability, we propose a pre-trained regularization strategy. Concretely, we regularize the distribution of each re-trained class anchor $\mathbf{w}^{\mathrm{r}}_{k^{\mathrm{r}}}$ to align more closely with that of its corresponding pre-trained anchor $\mathbf{w}^{\mathbb{p}}_{k^{\mathbb{p}*}(k^{\mathrm{r}})}$ as:

$$\mathcal{L}_{reg} = -\lambda \sum_{k^{\mathrm{r}} \in [K^{\mathrm{r}}]} \alpha_{k^{\mathrm{r}}} \frac{\mathbf{w}^{\mathbb{p}}_{k^{\mathbb{p}*}(k^{\mathrm{r}})} \cdot \mathbf{w}^{\mathrm{r}}_{k^{\mathrm{r}}}}{\left\| \mathbf{w}^{\mathbb{p}}_{k^{\mathbb{p}*}(k^{\mathrm{r}})} \right\|_2 \left\| \mathbf{w}^{\mathrm{r}}_{k^{\mathrm{r}}} \right\|_2}, \tag{4}$$

where $\alpha_{k^{\mathrm{r}}} \in [0, 1]$ denotes the per-class regularization weights and $\lambda$ controls the overall regularization strength. The weights $\alpha_{k^{\mathrm{r}}}$ are flexible and can incorporate various re-weighting schemes. Here, we simply use a cardinality-driven function inspired by Zhong et al. (2021), *i.e.*, $\alpha_{k^{\mathrm{r}}} = f(|\mathcal{T}_{k^{\mathrm{r}}}|) = 1 - \frac{|\mathcal{T}_{k^{\mathrm{r}}}| - \min |\mathcal{T}_{k^{\mathrm{r}}}|}{\max |\mathcal{T}_{k^{\mathrm{r}}}| - \min |\mathcal{T}_{k^{\mathrm{r}}}|}$, with weights normalized from 0 for head classes to 1 for tail classes.

## 5 EXPERIMENTS

We evaluate the `PTClf` paradigm from the following perspective. First, we conduct extensive ablation studies to validate the effectiveness of the paradigm itself. Second, we perform comprehensive benchmark evaluations on multiple long-tailed datasets to demonstrate the superiority of the paradigm across various methods. Third, we provide an in-depth analysis to examine the extensibility of the pre-trained classifier across diverse settings.

### 5.1 EXPERIMENTAL SETUPS

We evaluate `PTClf` on a series of datasets grouped into three categories: (1) **Long-tailed learning** encompasses 8 benchmark tasks, including 3 common long-tailed datasets: CIFAR100-LT, Places-LT, and iNaturalist 2018, as well as 5 newly introduced fine-grained visual classification (FGVC) tasks: FGVC Aircraft, CUB-200-2011, Oxford Flowers, Stanford Cars, and Stanford Dogs, each adapted to the long-tailed setting. (2) **Few-shot learning** includes the same 5 fine-grained datasets, each adapted to the few-shot setting. (3) **Domain generalization** includes 2 generalization tasks: CIFAR10-LT $\rightarrow$ CIFAR10.1 and STL10, as well as ImageNet-LT $\rightarrow$ ImageNet-V2, -Sketch, -A, and -R. For long-tailed datasets, in addition to evaluating overall accuracy, we follow the protocol in (Liu et al., 2019) and further report accuracy on three class splits: Head classes ($>100$ images), Median classes ($20\sim100$ images), and Tail classes ($<20$ images).

### 5.2 ABLATION STUDIES

**`PTClf` components (PTInit and PTReg) demonstrate strong effectiveness.** We conduct an ablative analysis of the initialization and regularization components in `PTClf`. Specifically, we design several contenders for the two components:

**Feat. Init.** (Init.) initializes the re-trained classifier with feature prototypes (Shi et al., 2024a). Similar to **PTInit**, this contender also exploits upstream knowledge for classifier initialization. However, it relies on the knowledge embedded in the pre-trained backbone instead of directly using the pre-trained classifier.

**LP-FT** (Init.) first trains a classifier via linear probing, and then uses it to initialize the re-trained classifier (Kumar et al., 2022). This contender is a representative approach in fine-tuning, aiming to mitigate feature distortion and preserve the upstream knowledge embedded in the pre-trained backbone.

**ETF** (Reg.) regularizes the re-trained classifier toward a simplex equiangular tight frame (Li et al., 2022). Similar to **PTReg**, this contender also constrains the re-trained class anchors toward better-structured anchors. However, it relies on randomly initialized simplex-frame anchors instead of leveraging informative pre-trained anchors.

**WB** (Reg.) regularizes excessively large norms in the re-trained classifier during fine-tuning (Alshammari et al., 2022). This contender is a representative approach in long-tailed learning, aiming to balance the classifier's norms.

As shown in Tab. 1, both **PTInit** and **PTReg** bring clear performance gains over the baseline, particularly on tail classes, and their combination leads to further improvements. More importantly, they substantially outperform their contenders, even those designed with similar goals (*i.e.*, *c1* for Init. and Reg.), highlighting the importance of leveraging the additional information within the pre-trained classifier. Fig. 6 further illustrates the channel behavior after integrating the components (see

Table 1: **Ablation study on** `PTClf` **components.** Results are presented in % using ViT-B/16 pre-trained on IN-21k. The baseline here is the LA loss (Menon et al., 2021) within Adapter+ (Steitz & Roth, 2024). [†]FGVCAircraft has no 'Head' classes, as its largest class contains only 67 samples.

| Method | Init. | Reg. | CIFAR100-IR200 | | | Places-LT | | | FGVCAircraft-LT | | |
|---|---|---|---|---|---|---|---|---|---|---|---|
| | | | All | Head | Tail | All | Head | Tail | All | Med[†] | Tail |
| Baseline | ✗ | ✗ | 85.2 | 92.4 | 77.4 | 45.9 | 48.1 | 41.6 | 30.7 | 57.6 | 18.9 |
| + PTInit | ✓ | ✗ | 87.2 ↑2.0 | 92.7 ↑0.3 | 81.9 ↑4.5 | 46.7 ↑0.8 | 48.0 ↓0.1 | 44.1 ↑2.5 | 32.3 ↑1.6 | 58.4 ↑0.8 | 20.8 ↑1.9 |
| *c1*. Feat. Init. (Shi et al., 2024a) | ✓ | ✗ | 84.2 ↓1.0 | 92.8 ↑0.4 | 73.9 ↓3.5 | 46.2 ↑0.3 | 48.3 ↑0.2 | 40.8 ↓0.8 | 31.3 ↑0.6 | 56.2 ↓1.4 | 20.2 ↑1.3 |
| *c2*. LP-FT (Kumar et al., 2022) | ✓ | ✗ | 83.1 ↓2.1 | 92.8 ↑0.4 | 72.6 ↓4.8 | 45.1 ↓0.8 | 48.2 ↑0.1 | 39.3 ↓2.3 | 28.9 ↓1.8 | 54.5 ↓3.9 | 17.6 ↓1.3 |
| + PTReg | ✗ | ✓ | 87.5 ↑2.3 | 92.7 ↑0.3 | 83.2 ↑5.8 | 47.7 ↑1.8 | 48.5 ↑0.4 | 45.6 ↑4.0 | 31.9 ↑1.2 | 58.6 ↑1.0 | 20.1 ↑1.2 |
| *c1*. ETF (Li et al., 2022) | ✗ | ✓ | 85.5 ↑0.3 | 92.7 ↑0.3 | 78.2 ↑0.8 | 46.0 ↑0.1 | 48.1 -0.0 | 41.9 ↑0.3 | 31.0 ↑0.3 | 57.8 ↑0.2 | 19.2 ↑0.3 |
| *c2*. WB (Alshammari et al., 2022) | ✗ | ✓ | 86.5 ↑1.3 | 89.7 ↓2.7 | 81.4 ↑4.0 | 45.1 ↓0.8 | 36.2 ↓11.9 | **49.5** ↑7.9 | 30.6 ↓0.1 | 44.9 ↓12.7 | **24.2** ↑5.3 |
| + PTInit + PTReg (Our `PTClf`) | ✓ | ✓ | **88.1** ↑2.9 | **92.8** ↑0.4 | **84.1** ↑6.7 | **47.9** ↑2.0 | **48.6** ↑0.5 | 45.8 ↑4.2 | **32.7** ↑2.0 | **58.8** ↑1.2 | 21.1 ↑2.2 |

*c1/c2*: Contender for initialization and regularization components.

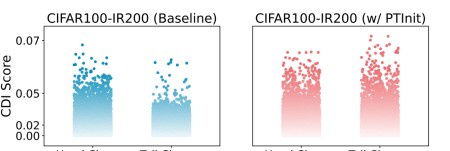 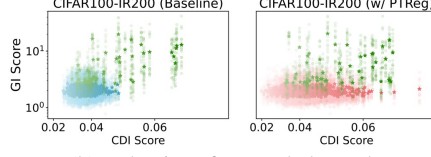

(a) Behavior of class-discriminative channels.     (b) Behavior of general channels.

Figure 6: **Ablation analysis of channel behavior.** **PTInit** and **PTReg** effectively guide the re-trained classifier toward oracle behavior in Figs. 3 and 4. Further analyses are in Appendix E.

Appendix E.1 for details). The re-trained classifier with **PTInit** effectively inherits the pre-trained class-discriminative channels, leading to a more balanced discriminative ability between head and tail classes. In addition, **PTReg** guides the re-trained classifier toward closer alignment with the oracle structure depicted in Fig. 3, thereby enhancing semantic awareness for tail classes.

**The only hyper-parameter $\lambda$ in `PTClf` is significantly robust.** `PTClf` introduces only one hyper-parameter, $\lambda$, which controls the regularization strength in Eq. (4). We investigate the impact of this parameter in Fig. 7. As shown, `PTClf` exhibits consistent gains (>1.5 pp) across a broad range of $\lambda$ values and datasets, demonstrating remarkable robustness. The best performance on all three datasets is attained at $\lambda \in [2.0, 3.0]$. For simplicity, we fix $\lambda = 2.0$ in most of our experiments.

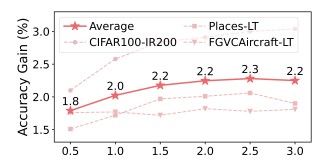

Figure 7: **Performance gains of `PTClf` across various $\lambda$ values.**

### 5.3 BENCHMARK RESULTS

**`PTClf` consistently improves performance across various LTL strategies.** We conduct comprehensive evaluations on the long-tailed benchmark across seven representative LTL strategies, w/ or w/o `PTClf`, including: data augmentation DODA (Wang et al., 2024), two-stage learning cRT (Kang et al., 2020) and LOS (Sun et al., 2025), balanced loss LA (Menon et al., 2021), supervised contrastive learning PaCo (Cui et al., 2021), ensemble learning ResLT (Cui et al., 2022), and the state-of-the-art foundation model-based approach LPT (Dong et al., 2023).

As shown in Tab. 2, applying prior LTL methods within the fine-tuning paradigm effectively achieves a more balanced performance compared to naïve CE training. Nevertheless, by simply leveraging the additional information within the pre-trained classifier, our proposed `PTClf` further enhances the performance of all strategies, with particularly remarkable improvements on tail classes. Moreover, in a fair comparison, where both methods adopt the same LTL tricks, `PTClf` outperforms the state-of-the-art foundation model-based approach LPT.

**`PTClf` consistently improves performance across various PEFT approaches.** Our proposed `PTClf` paradigm is broadly applicable to different fine-tuning methods. Therefore, we also conduct evaluations on the long-tailed benchmark across three representative PEFT in Appendix E.2.

### 5.4 IN-DEPTH ANALYSIS

**RQ1:** *Although the channel behavior appears closer to the oracle, is it genuinely so?* We evaluate the cosine similarity between the re-trained classifier trained on CIFAR100-IR200 and the oracle

Table 2: **Benchmark Results across various LTL strategies.** Results are presented in % using ViT-B/16 pre-trained on IN-21k. The baseline here is Adapter+ (Steitz & Roth, 2024) with different LTL strategies. [†]Most FGVC datasets have no 'Head' classes, thereby reporting accuracy for 'Median' classes. [‡]For a fair comparison, we apply the same LTL tricks used in LPT, *i.e.*, cosine classifier, mixup, deferred re-weighting, and balanced loss, and also reproduce the results on Places-LT.

| Method | C100-IR200 | | | C100-IR100 | | | C100-IR50 | | | Places-LT | | | iNaturalist | | | FGVC-LT | | |
|---|---|---|---|---|---|---|---|---|---|---|---|---|---|---|---|---|---|---|
| | A | H | T | A | H | T | A | H | T | A | H | T | A | H | T | A | M[†] | T |
| CE | 77.5 | 95.1 | 57.1 | 82.6 | 94.6 | 66.8 | 86.6 | 93.8 | 76.0 | 38.1 | 52.1 | 21.3 | 72.8 | 79.2 | 69.2 | 54.8 | 79.6 | 43.7 |
| + PTClf | **82.6** | **95.2** | **69.4** | **85.8** | **94.8** | **75.1** | **88.1** | **94.0** | **80.6** | **40.6** | **52.8** | **26.9** | **74.2** | **79.3** | **74.0** | **63.5** | **81.6** | **54.8** |
| DODA (Wang et al., 2024) | 85.6 | 93.2 | 77.8 | 88.1 | 92.7 | 83.2 | 89.7 | 92.2 | 88.1 | 44.8 | 47.9 | 39.7 | 78.2 | 73.3 | 78.6 | 62.9 | 77.4 | 55.6 |
| + PTClf | **88.6** | **93.3** | **84.4** | **90.0** | **92.8** | **87.3** | **90.8** | **92.4** | **90.7** | **47.3** | **48.2** | **45.0** | **79.4** | **74.1** | **80.5** | **67.9** | **79.7** | **62.0** |
| cRT (Kang et al., 2020) | 85.0 | **91.1** | 79.4 | 86.7 | 90.5 | 83.5 | 88.2 | 90.8 | 86.9 | 40.6 | **39.9** | 40.1 | 76.4 | 73.0 | 79.2 | 65.3 | 79.4 | 57.7 |
| + PTClf | **87.4** | 89.6 | **85.7** | **89.4** | **90.6** | **88.8** | **90.2** | **90.9** | **90.9** | **44.7** | 38.2 | **48.3** | **77.2** | **73.2** | **81.1** | **69.9** | **80.7** | **64.3** |
| LOS (Sun et al., 2025) | 61.5 | **95.5** | 22.7 | 67.8 | **95.3** | 26.4 | 74.2 | **93.8** | 32.7 | 22.9 | **46.7** | 0.4 | 59.4 | **70.6** | 48.4 | 18.6 | **45.6** | 9.7 |
| + PTClf | **73.3** | 72.2 | **77.4** | **73.9** | 71.1 | **80.5** | **74.3** | 73.4 | **79.4** | **33.1** | 31.7 | **33.5** | **68.9** | 66.2 | **70.4** | **33.1** | 29.8 | **34.5** |
| LA (Menon et al., 2021) | 85.2 | 92.4 | 77.4 | 87.9 | 92.3 | 83.2 | 89.8 | 92.1 | 88.8 | 45.9 | 48.1 | 41.6 | 78.6 | 73.0 | 79.4 | 58.2 | 74.0 | 50.5 |
| + PTClf | **88.1** | **92.8** | **84.1** | **89.8** | **92.4** | **87.3** | **90.6** | **92.2** | **90.4** | **47.9** | **48.6** | **45.8** | **79.4** | **73.9** | **80.6** | **65.1** | **76.6** | **59.2** |
| PaCo (Cui et al., 2021) | 85.8 | **82.5** | 86.6 | 87.5 | **84.1** | 90.2 | 88.8 | **85.8** | 93.4 | 45.2 | 41.5 | 44.1 | 78.1 | 67.7 | 79.8 | 65.1 | 68.5 | 61.6 |
| + PTClf | **86.0** | 81.1 | **88.2** | **87.7** | 82.5 | **91.8** | **88.9** | 85.5 | **94.1** | **45.7** | **41.6** | **45.1** | **78.5** | **67.9** | **80.4** | **67.3** | **69.1** | **64.8** |
| ResLT (Cui et al., 2022) | 88.9 | 92.1 | 85.4 | 90.5 | **91.5** | 89.2 | 91.4 | **91.6** | 93.2 | 48.4 | 38.9 | 50.8 | 80.2 | 71.8 | 82.0 | 67.9 | 79.0 | 61.5 |
| + PTClf | **89.6** | **92.2** | **87.1** | **91.0** | 91.2 | **91.0** | **91.5** | 91.1 | **94.2** | **49.4** | **39.2** | **53.0** | **80.6** | **72.1** | **82.4** | **70.7** | **80.0** | **65.5** |
| LPT (Dong et al., 2023)[‡] | 87.9 | - | - | 89.1 | - | - | 90.0 | - | - | 49.7 | 47.6 | 48.4 | 76.1 | - | 79.3 | - | - | - |
| PTClf[‡] | **88.7** | **91.3** | **86.6** | **89.8** | **90.4** | **88.9** | **90.6** | **90.5** | **92.4** | **50.1** | **49.8** | **48.5** | **80.2** | **74.3** | **81.6** | **73.6** | **83.6** | **68.2** |

**A**: All; **H**: Head; **M**: Median; **T**: Tail; **C100**: CIFAR100.

classifier trained on CIFAR100. As shown in Fig. 8, PTClf, especially the **PTReg** component, indeed brings the re-trained classifier much closer to the oracle classifier.

> **Insight 1:** The pre-trained classifier can effectively guide the re-trained classifier toward the oracle one.

**RQ2:** *How does the knowledge capacity of the pre-trained classifier affect its effectiveness?* Intuitively, the knowledge capacity of the pre-trained classifier, *i.e.*, the number of pre-trained classes $K^{\mathbb{p}}$, may impact its effectiveness. To investigate this, we evaluate models with different capacities (pre-trained on IN-21k and -1k) and further vary them by progressively pruning the pre-trained class anchors from 0% to 80% on CIFAR100-IR200. As shown in Fig. 9, a larger capacity tends to bring greater performance gains; however, even the smallest-capacity model (pre-trained on IN-1k) still delivers improvements.

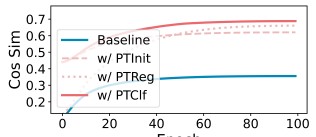

Figure 8: **Distance between the re-trained and oracle classifiers.**

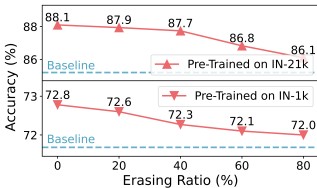

Figure 9: **Impact of classifier's knowledge capacity.**

> **Insight 2:** The pre-trained classifier with higher capacity tends to yield larger improvements. Moreover, even the lowest-capacity classifier (pre-trained on IN-1k) still delivers benefits.

**RQ3:** *Is the pre-trained classifier still useful when the downstream task is out-of-distribution (OOD) relative to the upstream task?* One might be skeptical that the pre-trained classifier benefits the downstream task because it has already been exposed to the upstream data. To examine this, we further evaluate on SVHN and DTD, which are OOD *w.r.t.* the upstream IN-21k dataset (see Appendix D.1 for dataset details). As shown in Tab. 3, the improvements from the pre-trained classifier are indeed less pronounced when such OOD datasets are class-balanced and data-rich. Nevertheless, when the OOD datasets are imbalanced, the pre-trained classifier still delivers substantial performance gains for the data-scarce tail classes.

Table 3: **Performance on OOD downstream tasks.**

| Method | S-Bal | S-IR20 | S-IR100 | S-IR500 |
|---|---|---|---|---|
| Baseline | 94.4 | 93.6 | 92.4 | 87.9 |
| + PTClf | **94.6** | **93.9** | **92.9** | **88.6** |
| Δ | ↑0.2 | ↑0.3 | ↑0.5 | ↑0.7 |

| Method | D-Bal | D-IR10 | D-IR20 | D-IR40 |
|---|---|---|---|---|
| Baseline | 76.4 | 62.7 | 58.3 | 49.2 |
| + PTClf | **77.1** | **63.6** | **60.4** | **55.2** |
| Δ | ↑0.7 | ↑0.9 | ↑2.1 | ↑6.0 |

**S**: SVHN; **D**: DTD.

> **Insight 3:** The pre-trained classifier is still significantly helpful even when the downstream task is out-of-distribution from the upstream task.

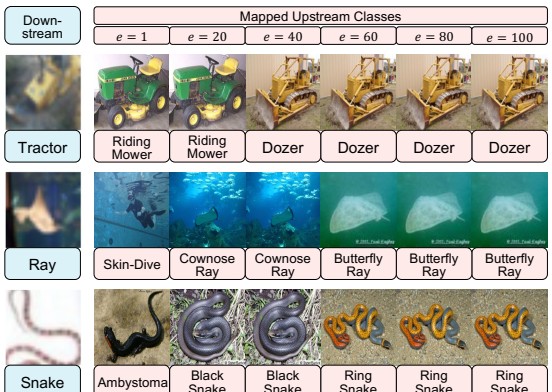

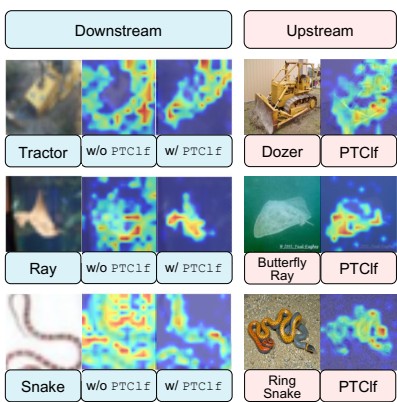

Figure 11: **Training dynamics of label mapping.** Each downstream class (left) is progressively aligned to its corresponding upstream class (right) as training proceeds.

Figure 12: **GradCAM visualization** of downstream classes (left) and their mapped upstream counterparts (right).

**RQ4:** *Does the pre-trained classifier benefit few-shot scenarios?* We conduct evaluations on the few-shot learning benchmark FGVC-FS. As illustrated in Fig. 10, the pre-trained classifier provides clear gains in learning with few-shot data, especially under severe data scarcity, underscoring the importance of leveraging additional information in such data-scarce scenarios. A detailed analysis is provided in Appendix C.

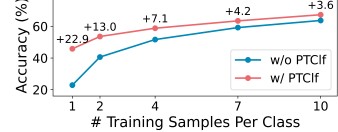

Figure 10: **Performance on few-shot learning.**

> **Insight 4:** The pre-trained classifier offers substantial benefits under data scarcity, achieving significant improvements in few-shot learning.

**RQ5:** *Does the pre-trained classifier benefit domain generalization?* We conduct evaluations on domain generalization tasks and compare `PTClf` (initialized with the pre-trained classifier) with a related DG approach, LP-FT (initialized with the linear-probed classifier), under the balanced LA loss. As shown in Tab. 4, the pre-trained classifier effectively preserves upstream knowledge and markedly improves robustness over the baseline. A detailed analysis is provided in Appendix C.

Table 4: **Performance on domain generalization.**

| Method | C10-LT | C10-DG | IN-LT | IN-DG |
|---|---|---|---|---|
| Baseline | 95.3 | 91.6 | 79.6 | 39.9 |
| LP-FT | 93.6 | 89.4 | 78.7 | 38.2 |
| + PTClf | **96.3** | **94.0** | **82.7** | **43.6** |
| Δ | ↑1.0 | ↑2.4 | ↑3.1 | ↑3.7 |

**C10**: CIFAR10; **DG**: Domain generalization.

> **Insight 5:** The pre-trained classifier is helpful for preserving the upstream knowledge and improving robustness, achieving significant improvements in domain generalization.

**RQ6:** *How does the pre-trained classifier help the re-trained classifier?* We provide additional visualizations to further elucidate the mechanisms of `PTClf`. Fig. 11 illustrates the evolution of label mapping during fine-tuning. As fine-tuning proceeds, the mapping progressively converges, with each downstream class aligning to a semantically relevant upstream class. Fig. 12 further presents GradCAM visualizations from models fine-tuned with and without `PTClf`, as well as from the pre-trained model. As shown, while the re-trained classifier tends to make erroneous predictions, the pre-trained classifier effectively guides it by transferring knowledge from relevant upstream classes, thereby improving classification performance.

> **Insight 6:** The pre-trained classifier is capable of guiding the re-trained classifier through knowledge transfer from relevant upstream classes.

## 6 CONCLUSION

In this paper, we investigate the re-trained and pre-trained classifiers in LTL. We observe that the naïvely re-trained classifier struggles on tail classes, exhibiting weakened discriminative ability and semantic awareness. By contrast, the often-overlooked pre-trained classifier behaves closely aligned with the oracle. Motivated by this, we propose `PTClf`, a novel fine-tuning paradigm that harnesses the pre-trained classifier to guide the re-trained one. Extensive experiments validate the effectiveness of `PTClf` in LTL while offering insights into the untapped potential of the pre-trained classifier.

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

# Appendix

This appendix presents further details and results that could not be included in the main paper due to space constraints. The content is organized as follows:

- § A provides detailed descriptions and validations of the channel-wise metrics, *i.e.*, CDI and GI.
- § B presents the comprehensive observations on the re-trained and pre-trained classifiers.
- § C presents the comprehensive insights on the pre-trained classifier.
- § D contains more details on the datasets, baselines, and implementation details.
- § E contains the full results corresponding to those reported in the main paper, as well as additional experiments.
- § F discusses the limitation of `PTClf` and points out the potential direction for our future work.
- § G describes the usage of large language models.

## A   METRIC VALIDATION

### A.1   DETAILED DESCRIPTIONS OF CDI AND GI

**Motivation for Channel-Level Analysis.** A key challenge in LTL is the severe imbalance in classifiers (Kang et al., 2020; Sun et al., 2025). Most existing studies examine this issue from a norm-level perspective, *i.e.*, considering the overall norms across classifier channels. Nevertheless, recent research has revealed that individual channels encode distinct information and contribute differently across classes (Zhu et al., 2023b; Zhao et al., 2024a; Zhang et al., 2024). Motivated by this, we perform a fine-grained, channel-level analysis to gain deeper insights.

**Class-Discriminative Importance (CDI).** CDI aims to identify channels that are important for discriminating a given class. We define the CDI score of channel $d$ for class $k^r$ as:

$$\text{CDI}_{k^r,d} = \left| w^r_{k^r,d} \right|. \tag{i}$$

Specifically, CDI has an intuitive interpretation: in the linear classifier $(\mathbf{w}^r_{k^r}{}^\top \mathbf{z}_i + b_{k^r})$, coefficients $w^r_{k^r,d}$ with larger magnitudes often indicate greater discriminative importance for class $k^r$. This observation has been widely validated in both statistics (Yuan & Lin, 2006; Li et al., 2021) and pruning (Ye et al., 2018; Chan & Veas, 2024) communities. Therefore, a high $\text{CDI}_{k^r,d}$ score indicates that channel $d$ serves as a strong discriminative factor for class $k^r$.

In addition, we also define the CDI score for a class set $\mathcal{K}^r \subseteq [K^r]$ as:

$$\text{CDI}_{\mathcal{K}^r,d} = \max_{k^r \in \mathcal{K}^r} \text{CDI}_{k^r,d}. \tag{ii}$$

Specifically, the max operation ensures that a channel is considered class-discriminative if it contributes strongly to any class within the set. We use this CDI score to visualize feature-channel behavior, for instance, in Figs. 4 and 6b.

**General Importance (GI).** GI aims to identify channels that are important for all classes. We define the GI score of channel $d$ as:

$$\text{GI}_d = \frac{1}{K^r} \sum_{k^r \in [K^r]} \frac{1}{|\mathcal{T}_{k^r}|} \left\| \mathbf{Z}_{\mathcal{T}_{k^r},d} \right\|_1. \tag{iii}$$

Specifically, GI builds upon the observations noted in (Zhao et al., 2024a): feature channels $\mathbf{Z}_{\mathcal{T}_{k^r},d} \in \mathbb{R}^{|\mathcal{T}_{k^r}|}$ with high norms tend to be broadly informative across different classes. Similar findings have also been reported in the pruning literature (Li et al., 2017). Therefore, a high $\text{GI}_d$ score indicates that channel $d$ is broadly important across different classes.

### A.2   VALIDATION OF CDI AND GI

We validate the effectiveness of CDI and GI from the following two perspectives:

**Accuracy-based Validation.** We gradually erase high-CDI or high-GI channels from the model fine-tuned on CIFAR100 and track the resulting changes in accuracy. Specifically, we randomly select 20 classes from CIFAR100 to compute CDI (on the 20 classes), GI (on all 100 classes), and

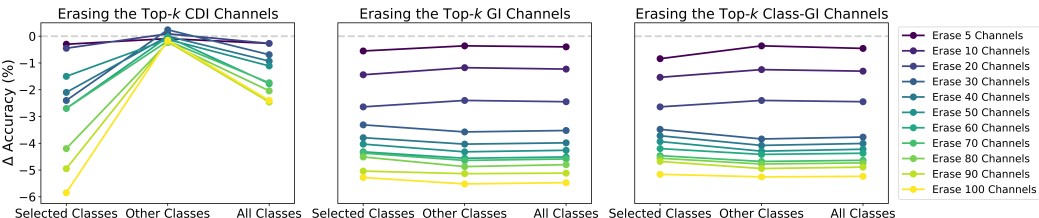

Figure i: **Impact of erasing different channels on accuracy.**

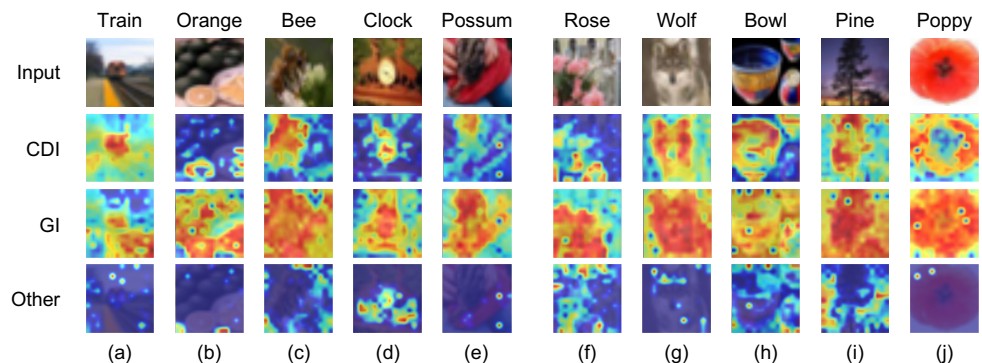

Figure ii: **GradCAM visualization of CDI and GI.**

class-GI (on the 20 classes). Regarding CDI, as shown in the left side of Fig. i, erasing high-CDI channels derived from the 20 selected classes leads to accuracy drops only within those classes, while the other 80 classes remain unaffected. This confirms that CDI effectively identifies channels specific to the given classes. Regarding GI, as shown in the middle of Fig. i, erasing high-GI channels derived from all 100 classes causes accuracy drops in all classes. This confirms that GI indeed captures channels shared across all classes. Additionally, the same global drops occur when erasing high-class-GI channels derived from only the 20 selected classes in the right side of Fig. i, indicating that the global effect of GI is not due to computing norms over all classes.

**Visualization-based Validation.** We also visualize class activation maps aggregated from the top-100 class-discriminative and general channels on CIFAR100 using GradCAM (Selvaraju et al., 2017). Two interesting observations emerge: **(1)** As shown in Fig. ii(a-e), activation maps from high-GI channels tend to highlight all objects relevant to the downstream task. For instance, in Fig. ii(a), high-GI channels attend not only to the train (the query class) but also to the road (another class in CIFAR100); similarly, in Fig. ii(c), high-GI channels highlight both the bee (the query class) and the flower (another CIFAR100 class). Simply put, when two conflicting task-relevant objects appear in the same image, high-GI channels tend to highlight both. In contrast, high-CDI channels focus more precisely on the object of the query class. **(2)** As shown in Fig. ii(f-j), high-CDI channels focus on class-specific regions, such as distinctive attributes of the query class, while high-GI channels concentrate on broader regions, such as entire foreground objects. For instance, in Fig. ii(g), high-GI channels attend to the whole wolf, whereas high-CDI channels identify specific attributes like the wolf's face. Overall, these observations confirm that high-CDI and high-GI channels indeed exhibit class-discriminative and general characteristics, respectively.

**Roles of CDI and GI.** Regarding CDI, our validation shows that it effectively identifies channels responsible for distinguishing the given classes. Therefore, we adopt an intuitive approach that uses the quantity of high-CDI channels (termed class-discriminative channels) as a straightforward measure of the classifier's discriminative ability for those given classes. Regarding GI, our validation shows that it instead captures channels associated with generic factors common across all classes. Therefore, we apply GI to investigate the semantic properties of the class-discriminative channels, *e.g.*, by examining whether these discriminative channels correspond to high-GI channels (termed general channels), thereby assessing the classifier's semantic awareness.

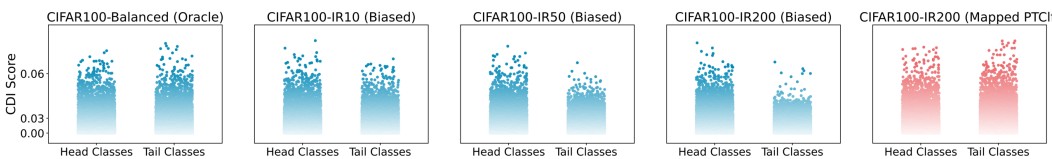

Figure iii: **Behavior of classifiers' class-discriminative channels** on CIFAR100 using CE loss.

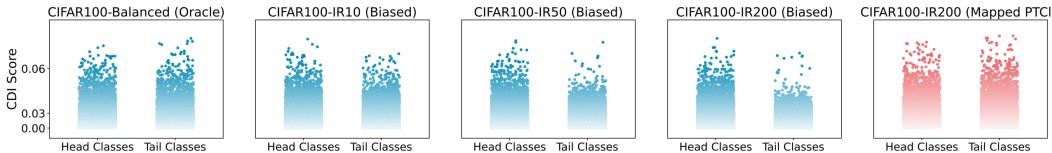

Figure iv: **Behavior of classifiers' class-discriminative channels** on CIFAR100 using LA loss.

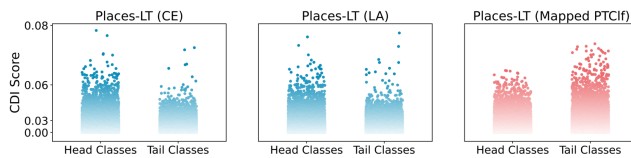

Figure v: **Behavior of classifiers' class-discriminative channels** on Places-LT.

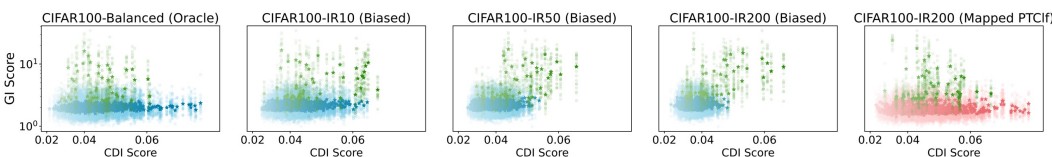

Figure vi: **Behavior of feature channels corresponding to class-discriminative channels** on CIFAR100 using CE loss.

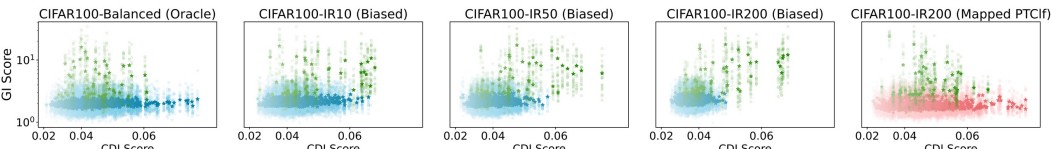

Figure vii: **Behavior of feature channels corresponding to class-discriminative channels** on CIFAR100 using LA loss.

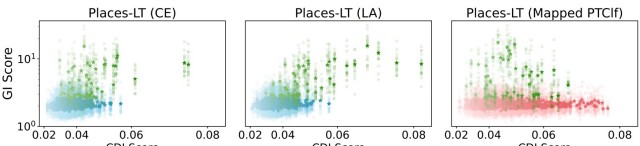

Figure viii: **Behavior of feature channels corresponding to class-discriminative channels** on Places-LT.

## B COMPREHENSIVE OBSERVATIONS ON RE-TRAINED CLASSIFIERS

We present a comprehensive analysis of classifiers in LTL with foundation models. In particular, we examine the re-trained classifier of the biased version (trained on the balanced full dataset) and the oracle version (trained on an imbalanced data subset), as well as the pre-trained classifier. Our main observations are as follows:

**Class-Discriminative Channel Imbalance.** In Figs. iii, iv, and v, we visualize the CDI scores of head- and tail-class anchors in the classifiers. As shown in Fig. iii, the class-discriminative channels of tail classes are highly sparse in the biased re-trained classifier, with this sparsity increasing as the imbalance ratio grows. In contrast, the class-discriminative channels for head classes remain consistently dense under both balanced and long-tailed settings. Moreover, Fig. iv demonstrates

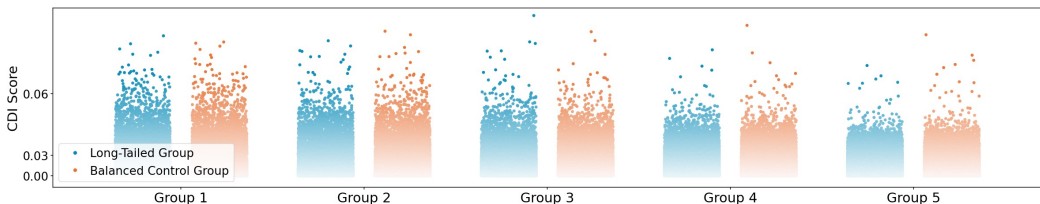

Figure ix: **Behavior of classifiers' class-discriminative channels** on long-tailed group and balanced control group.

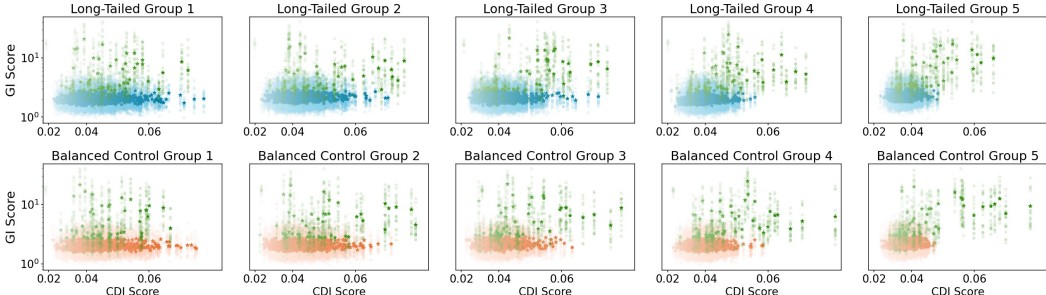

Figure x: **Behavior of feature channels corresponding to class-discriminative channels** on long-tailed group and balanced control group.

that this pattern remains evident even with the balanced LA loss (Menon et al., 2021), while Fig. v shows that the same sparsity in tail classes appears on Places-LT. This channel imbalance explicitly demonstrates that the re-trained classifier captures far fewer discriminative channels for tail classes than head classes, resulting in a significantly weaker discriminative ability for tail classes. This leads to our first observation:

> **Observation 1:** The re-trained classifier exhibits an imbalance in the class-discriminative channels, in which those associated with tail classes are highly sparse. This leads to the classifier's discriminative ability being far less effective for tail classes than for head classes.

**Class-Discriminative Channel Mislearning.** In Figs. vi, vii, and viii, we further visualize the GI scores of high-CDI channels in tail-class anchors. As shown in Fig. vi, we observe that the class-discriminative channels in the biased re-trained classifier increasingly rely on general channels as the imbalance ratio grows. This stands in sharp contrast to the oracle classifier, which predominantly leverages less-general channels as class-specific channels. Moreover, Fig. vii demonstrates that this pattern persists even with the balanced LA loss (Menon et al., 2021), while Fig. viii shows that the same misreliance on general features appears on Places-LT. This channel misreliance indicates that the re-trained classifier tends to mislearn general channels as the primary class-discriminative factors for tail classes, reflecting weak semantic awareness. This leads to our second observation:

> **Observation 2:** The re-trained classifier suffers from weakened semantic awareness for tail classes, resulting in class-discriminative channels being mislearned as depending exclusively on general channels.

**Root cause of both observations.** To investigate the underlying causes of the two issues above, we design a controlled experiment that separates the effects of data quantity and class imbalance. We first split the 100 classes of CIFAR100 into five groups of 20 classes, where Group 1 corresponds to head classes and Group 5 to tail classes. The *long-tailed groups* are trained on imbalanced data (CIFAR100-IR200) and evaluated on their corresponding 20-class test set. For each long-tailed group, we construct a *balanced control group* that matches the same average number of samples per class. For instance, if the long-tailed Group 5 has an average of 4 training samples across its selected 20 classes, then each class in its balanced control group is assigned exactly 4 training samples. The balanced control groups are also evaluated on the same 20-class test set.

As illustrated in Figs. ix and x, both channel imbalance and mislearning appear when comparing balanced control groups with varying sample sizes. Specifically:

- Fig. ix shows that the sparsity in class-discriminative channels increases as the number of training samples decreases. This indicates that data scarcity is the main cause of channel sparsity, since the re-trained classifier struggles to learn sufficient discriminative factors from limited data.
- Fig. x shows that class-discriminative channels increasingly rely on general features as the imbalance ratios grow. This suggests that data scarcity is also the main cause of channel mislearning, since the re-trained classifier fails to recognize class-specific (less-general) features from limited data, thereby leaving the tail classes that can only leverage the general features instead.

These findings clearly demonstrate that the main culprit of channel imbalance and mislearning in LTL is the scarcity of data in tail classes. This leads to our third observation:

> **Observation 3:** The scarcity of tail-class samples is the root cause of both weakened discriminative ability and semantic awareness.

## C COMPREHENSIVE INSIGHTS ON THE PRE-TRAINED CLASSIFIER

**RQ1:** *Although the channel behavior appears closer to the oracle, is it genuinely so?* We evaluate the cosine similarity between the re-trained classifier trained on the imbalance dataset CIFAR100-IR200 and the oracle classifier trained on the full set of CIFAR100. As shown in Fig. xi, `PTClf`, especially the **PTReg** component, indeed brings the re-trained classifier much closer to the oracle.

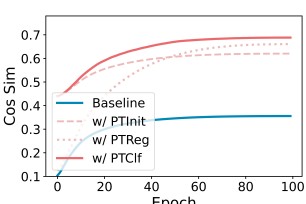

Figure xi: **Distance between the re-trained and oracle classifiers.**

> **Insight 1:** The pre-trained classifier can effectively guide the re-trained classifier toward the oracle one.

**RQ2:** *How does the knowledge capacity of the pre-trained classifier affect its effectiveness?* Intuitively, the knowledge capacity of the pre-trained classifier, *i.e.*, the number of upstream classes, may impact its effectiveness. To probe this, we evaluate models with different capacities and progressively prune the pre-trained class anchors from 0% to 80% on CIFAR100-IR200. As shown in Fig. xii, greater capacity tends to bring larger performance gains, whereas even the smallest-capacity model (pre-trained on IN-1k) still delivers improvements.

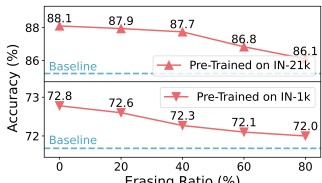

Figure xii: **Impact of classifier's knowledge capacity.**

> **Insight 2:** The pre-trained classifier with higher capacity tends to yield larger improvements; nonetheless, even the lowest-capacity classifier (pre-trained on IN-1k) still delivers benefits.

**RQ3:** *Is the pre-trained classifier still useful when the downstream task is out-of-distribution (OOD) relative to the upstream task?* One might be skeptical that the pre-trained classifier benefits the downstream task because it has already been exposed to the upstream data. To examine this, we further evaluate on SVHN and DTD, which are OOD *w.r.t.* the upstream IN-21k (Russakovsky et al., 2015). As shown in Tab. i, the improvements from the pre-trained classifier are indeed less pronounced when such OOD datasets are class-balanced and data-rich. Nevertheless, when the OOD datasets are imbalanced, the pre-trained classifier still delivers substantial performance gains for the data-scarce tail classes.

Table i: **Performance on OOD downstream tasks.**

| Method | S-Bal | S-IR20 | S-IR100 | S-IR500 |
|---|---|---|---|---|
| Baseline | 94.4 | 93.6 | 92.4 | 87.9 |
| + `PTClf` | **94.6** | **93.9** | **92.9** | **88.6** |
| Δ | ↑0.2 | ↑0.3 | ↑0.5 | ↑0.7 |

| Method | D-Bal | D-IR10 | D-IR20 | D-IR40 |
|---|---|---|---|---|
| Baseline | 76.4 | 62.7 | 58.3 | 49.2 |
| + `PTClf` | **77.1** | **63.6** | **60.4** | **55.2** |
| Δ | ↑0.7 | ↑0.9 | ↑2.1 | ↑6.0 |

**S**: SVHN; **D**: DTD.

> **Insight 3:** The pre-trained classifier is still significantly helpful even when the downstream task is out-of-distribution from the upstream task.

**RQ4:** *Does the pre-trained classifier benefit few-shot scenarios?* We conduct evaluations on the few-shot learning benchmark FGVC-FS. As illustrated in Fig. xiii, the pre-trained classifier provides clear gains in learning with few-shot data, especially under severe data scarcity (*i.e.*, only one

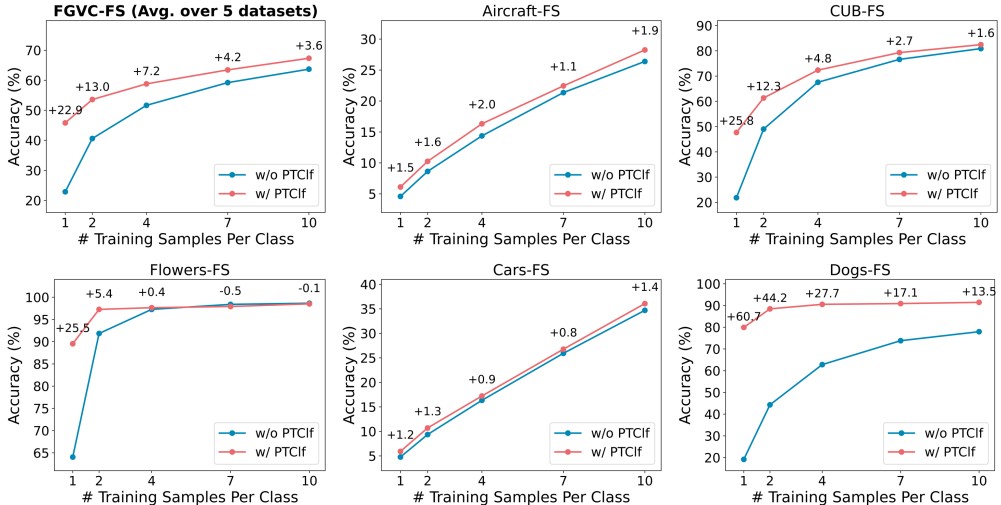

Figure xiii: **Performance on few-shot learning.**

Table ii: **Performance on domain generalization.**

| Method | Source | Target | | Source | Target | | | |
|---|---|---|---|---|---|---|---|---|
| | C10-LT | STL | C10.1 | IN-LT | IN-V2 | IN-S | IN-A | IN-R |
| Baseline | 95.3 | 91.6 | 91.7 | 79.6 | 21.7 | 34.0 | 35.0 | 68.9 |
| LP-FT (Kumar et al., 2022) | 93.6 | 89.4 | 89.5 | 78.7 | 19.0 | 32.5 | 33.4 | 67.8 |
| + PTClf | **96.3** | **94.1** | **94.0** | **82.7** | **24.6** | **38.3** | **39.5** | **72.0** |
| Δ | ↑2.0 | ↑2.5 | ↑2.3 | ↑3.1 | ↑2.9 | ↑4.3 | ↑4.5 | ↑3.1 |
| SWAD (Cha et al., 2021) | 91.8 | 85.2 | 88.3 | 80.1 | 22.5 | 35.2 | 35.9 | 69.6 |
| + PTClf | **95.7** | **89.9** | **92.3** | **82.9** | **24.4** | **39.4** | **39.8** | **72.8** |
| Δ | ↑3.9 | ↑4.7 | ↑4.0 | ↑2.8 | ↑1.9 | ↑4.2 | ↑3.9 | ↑3.2 |
| SPD (Tian et al., 2024) | 95.2 | 96.1 | 93.3 | 80.4 | 24.2 | 38.1 | 36.9 | 72.2 |
| + PTClf | **97.7** | **97.8** | **97.4** | **83.4** | **25.8** | **39.5** | **40.6** | **73.2** |
| Δ | ↑2.5 | ↑1.7 | ↑4.1 | ↑2.0 | ↑1.6 | ↑1.4 | ↑3.7 | ↑1.0 |

**C10**: CIFAR10; **C10.1**: CIFAR10.1; **DG**: Domain generalization.

training sample per class), underscoring the importance of leveraging additional information in such data-scarce scenarios.

> **Insight 4:** The pre-trained classifier offers substantial benefits under data scarcity, achieving significant improvements in few-shot learning.

**RQ5:** *Does the pre-trained classifier benefit domain generalization?* We conduct evaluations on domain generalization tasks and compare PTClf (initialized with the pre-trained classifier) with a related method, LP-FT (initialized with the linear-probed classifier), under the balanced LA loss. To further highlight its versatility, we also incorporate PTClf into two stronger DG baselines, SWAD (Cha et al., 2021) and SPD (Tian et al., 2024), both using the LA loss. As shown in Tab. ii, the pre-trained classifier effectively preserves upstream knowledge and markedly improves robustness across different baselines.

> **Insight 5:** The pre-trained classifier is helpful for preserving the upstream knowledge and improving robustness, achieving significant improvements in domain generalization.

**RQ6:** *How does the pre-trained classifier help the re-trained classifier?* We provide additional visualizations to further elucidate the mechanisms of PTClf. Fig. xiv illustrates the evolution of label mapping during fine-tuning. As fine-tuning progresses, the mapping progressively converges, with each downstream class aligning to a semantically similar upstream class. Fig. xv presents examples where the baseline (without PTClf) makes errors but PTClf correctly classifies at inference time. As shown, when the re-trained classifier fails to recognize the downstream classes, the pre-trained

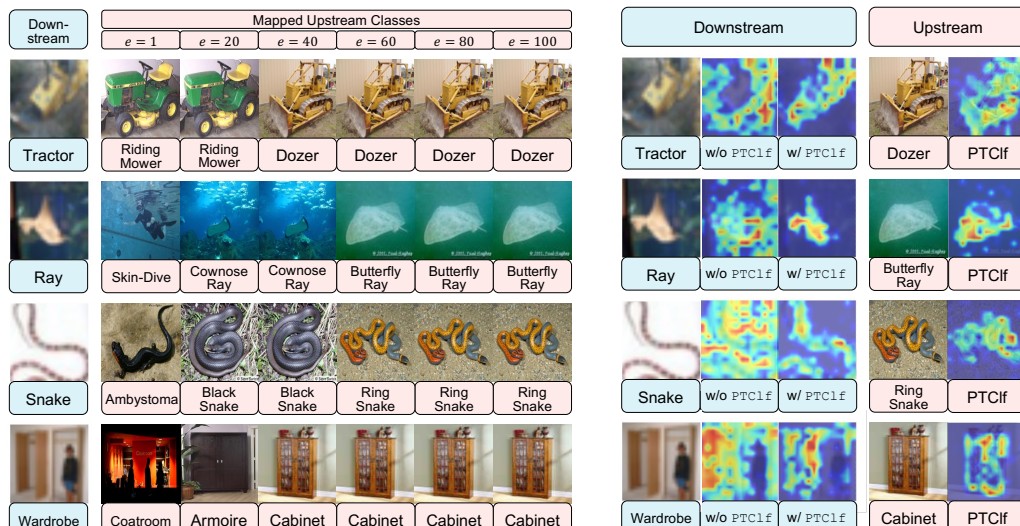

Figure xiv: **Training dynamics of label mapping.** Each downstream class (left) is progressively aligned to its corresponding upstream class (right) as training proceeds.

Figure xv: **GradCAM visualization** of downstream classes (left) and their mapped upstream counterparts (right).

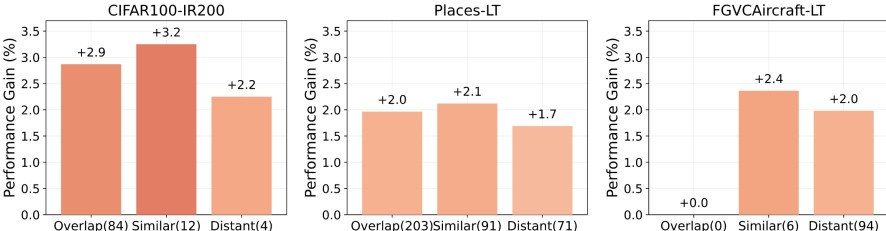

Figure xvi: **Performance gains across different levels of category overlap.**

classifier can effectively guide it by transferring knowledge from semantically relevant upstream classes, thereby improving classification by harnessing these additional priors.

> **Insight 6:** The pre-trained classifier is capable of guiding the re-trained classifier through knowledge transfer from semantically related upstream classes.

**RQ7:** *Does the utility of the pre-trained classifier stem mainly from class overlap?* A natural concern is that the benefits of the pre-trained classifier might stem largely from overlapping classes, rather than from genuine knowledge transfer among non-overlapping classes. To investigate this, we quantify the relatedness between upstream and downstream labels using WordNet path similarity and divide them into three groups:

- **Overlap:** Identical labels or hypernym-hyponym pairs (path similarity $\in [0.9, 1.0]$);
- **Similar:** Semantically related but not hierarchically identical categories (path similarity $\in [0.6, 0.9)$);
- **Distant:** Weakly related or unrelated categories (path similarity $\in [0.0, 0.6)$).

As shown in Fig. xvi, PTClf delivers consistent improvements across all three groups, covering both overlapping or non-overlapping classes. More importantly, the largest gains arise from the Similar group rather than the Overlap group, while the Distinct group also exhibits clear and comparable gains.

> **Insight 7:** The pre-trained classifier does benefit from overlapping classes, but its primary advantage stems from genuine knowledge transfer among non-overlapping classes.

**RQ8:** *Does the utility of the pre-trained classifier stem mainly from better classifier properties?* Apart from class overlap, another natural concern is that the benefits of the pre-trained classifier might instead stem from better classifier characteristics, such as more balanced channel distribution

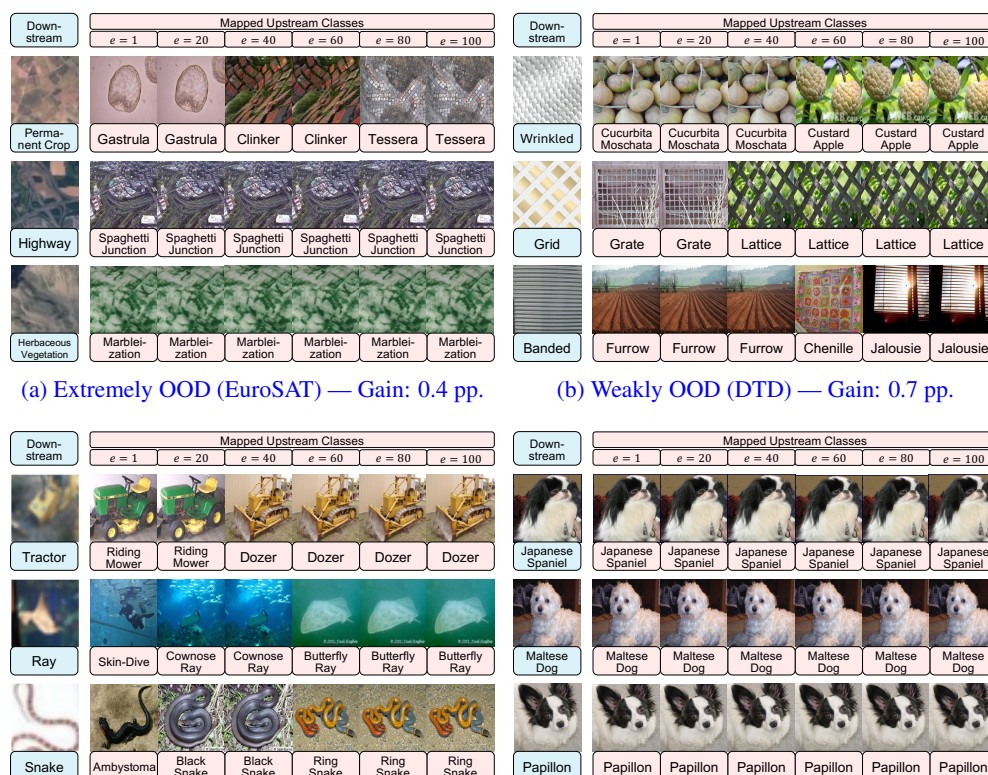

(a) Extremely OOD (EuroSAT) — Gain: 0.4 pp.  (b) Weakly OOD (DTD) — Gain: 0.7 pp.

(c) Moderate Overlap (CIFAR100) — Gain: 2.9 pp.  (d) Complete Overlap (Dogs) — Gain: 18.3 pp.

Figure xvii: Mapping dynamics across datasets from extremely OOD to completely ID.

or closer alignment with the feature backbone. To investigate this, we replace our label mapping with a random mapping, which retains the potential benefits from better classifier properties while explicitly removing knowledge correlations between classes. As shown in Tab. iii, performance consistently degrades under random mapping. This confirms that the observed gains cannot be attributed solely to distribution or alignment effects, but instead rely on correlated knowledge.

Table iii: **Performance under random mapping.**

| Method | C100-IR200 | Places-LT | Aircraft-LT |
|---|---|---|---|
| Baseline | 85.2 | 45.9 | 30.7 |
| Our Mapping | **88.1** | **47.9** | **32.7** |
| Random Mapping | 84.6 | 45.8 | 30.4 |
| △ | ↓0.6 | ↓0.1 | ↓0.3 |

**Insight 8:** The pre-trained classifier's benefits are not simply due to better classifier properties, but rather to the transfer of correlated knowledge.

**RQ9:** *Is knowledge transfer indeed the main factor behind PTClf's gains?* To examine the mechanism of knowledge transfer in PTClf, we further visualize the mapping dynamics across a series of datasets that range from extremely out-of-distribution (OOD) to completely in-distribution (ID):

• **EuroSAT (extremely OOD)** contains satellite imagery and is the most disjoint dataset relative to natural-image ImageNet. As shown in Fig. xviia, despite this large domain gap, the learned mapping still pairs each land-use class with ImageNet classes that share analogous global structure or layout (*e.g.*, downstream Herbaceous_Vegetation aligning with upstream Marbleization owing to similar veining patterns), thereby yielding a small but measurable improvement (∼0.4 pp).

• **DTD (weakly OOD)** consists of texture categories that have no direct counterparts in ImageNet. As shown in Fig. xviib, the mapping still converges to ImageNet classes with related semantic attributes (*e.g.*, downstream Banded aligning with upstream Jalousie owing to similar banded patterns), thereby yielding a modest improvement (∼0.7 pp).

• **CIFAR100 (moderate overlap)** contains natural images and shares partial label overlap within ImageNet. As shown in Fig. xviic, the mapping effectively aligns each downstream class to semantically relevant ImageNet classes (*e.g.*, downstream Tractor aligning with upstream Dozer based on similar overall shape), thereby yielding a notable gain (∼2.9 pp).

- **Stanford Dogs (complete overlap)** is a subset of ImageNet, where each downstream class corresponds exactly to a specific ImageNet class. As shown in Fig. xviid, the mapping can directly match the exact corresponding ImageNet label, thereby yielding a significant improvement (up to ~18.3 pp).

Taken together, these results lead to two key observations: **(1)** `PTClf`'s improvements scale with the knowledge correlation between upstream and downstream labels, with datasets showing stronger alignment (*e.g.*, CIFAR100 and Stanford Dogs) exhibiting larger gains. Such a clear trend effectively supports that knowledge transfer is indeed the main driver of `PTClf`'s performance. **(2)** Even for OOD datasets like EuroSAT and DTD, the learned mappings align downstream labels to disjoint yet semantically related upstream classes. This suggests that the essence of knowledge transfer lies in transferring semantically related knowledge, rather than simply reusing identical labels.

> **Insight 9:** The pre-trained classifier's benefits arise mainly from transferring knowledge among semantically related upstream classes.

**RQ10:** *Can the pre-trained classifier's benefits extend to the LLM domain?* To investigate whether the benefits of the pre-trained classifier extend beyond vision, we apply our method to the LLM domain on a token-level classification task. Specifically, we fine-tune BERT on the CoNLL2003 named entity recognition dataset. As shown in Tab. iv, `PTClf` delivers notable improvements in this LLM fine-tuning setting. This preliminary experiment suggests that the advantages of the pre-trained classifier are not limited to image classification, but also extend to LLM-based classification tasks.

Table iv: **Performance on CoNNL2003.**

| Method | CoNLL2003 |
|---|---|
| Baseline | 90.8 |
| + PTClf | **91.9** |

> **Insight 10:** The pre-trained classifier's benefits can be extended to LLM-based classification.

# D    MORE INFORMATION ON EXPERIMENTS

## D.1    DATASETS

We give an overview of the datasets used in our paper. The datasets can be grouped into:

- **Long-Tailed Learning.** The long-tailed benchmark aims to assess performance on real-world long-tailed data. Following prior works (Kang et al., 2020; Shi et al., 2024a;a), we include 3 widely used datasets: CIFAR100-LT, Places-LT, and iNaturalist 2018. In addition, we introduce 5 fine-grained visual classification (FGVC) tasks: FGVC Aircraft, CUB-200-2011, Oxford Flowers, Stanford Cars, and Stanford Dogs, aiming to further evaluate transferability. Each FGVC task is adapted to the long-tailed setting.
- **Few-Shot Learning.** The few-shot benchmark aims to assess performance in extreme data-scarce scenarios. We use the same 5 fine-grained datasets, each adapted to the few-shot setting.
- **Domain Generalization.** The domain generalization tasks aim to assess the model's ability to generalize across domains after fine-tuning on a specific one. We includes 2 generalization tasks: CIFAR10-LT → CIFAR10.1 and STL10, and ImageNet-LT → ImageNet-V2, -Sketch, -A, and -R. For CIFAR10, CIFAR10.1 (Recht et al., 2018) and STL (French et al., 2017) serve as standard domain adaptation tasks, where CIFAR10 is the in-distribution (ID) dataset and CIFAR10.1 and STL are the out-of-distribution (OOD) datasets. Since CIFAR10 contains no "monkey" class, we follow prior work (Kumar et al., 2022) and remove it from STL. For ImageNet-1K, we fine-tune on ImageNet-LT (ID) and evaluate on 4 widely used OOD datasets: ImageNet-V2 (Recht et al., 2019), ImageNet-Sketch (Wang et al., 2019), ImageNet-A (Hendrycks et al., 2021b), and ImageNet-R (Hendrycks et al., 2021a).
- **Completely OOD Datasets *w.r.t.* ImageNet.** The OOD datasets aim to assess the robustness of ImageNet-pretrained models on data that are entirely out-of-distribution from their pre-trained task. We specifically consider two datasets: SVHN and DTD. SVHN consists of digit images captured from real-world street scenes, which differ fundamentally from the object-centric categories in ImageNet. DTD contains texture-centric images annotated with human-describable attributes (*e.g.*, "striped", "zigzagged"), making it distributionally distinct from ImageNet's natural object classes. These characteristics ensure that both SVHN and DTD serve as strictly out-of-distribution datasets *w.r.t.* ImageNet.

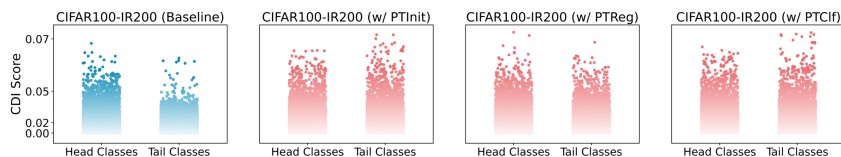

Figure xviii: **Behavior of classifiers' class-discriminative channels** on Places-LT.

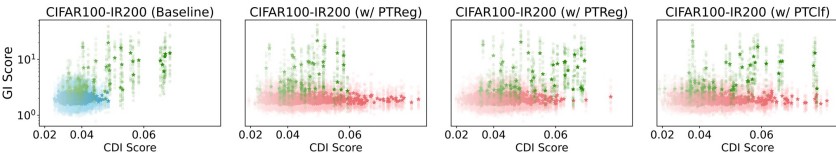

Figure xix: **Behavior of feature channels corresponding to class-discriminative channels** on CIFAR100 using CE loss.

## D.2 BASELINES

Our comparisons include two different types of baseline methods:

- **Traditional Long-Tailed Training Strategy.** We evaluate the effectiveness of integrating our paradigm with six representative long-tailed training strategies, including the data augmentation method DODA (Wang et al., 2024), the two-stage approaches cRT (Kang et al., 2020) and LOS (Sun et al., 2025), the balanced loss LA (Menon et al., 2021), the supervised contrastive learning PaCo (Cui et al., 2021), and the ensemble approach ResLT (Cui et al., 2022).
- **State-of-the-Art LTL with Foundation Models.** We also compare our paradigm with state-of-the-art long-tailed learning methods based on foundation models. To the best of our knowledge, LPT (Dong et al., 2023) is the *only* visual-only state-of-the-art fine-tuning approach. Accordingly, we benchmark our paradigm against LPT. Since LPT incorporates several long-tailed training tricks, such as cosine classifier, mixup, deferred re-weighting, and balanced loss, we also integrate these tricks when comparing with LPT to ensure fairness.

## D.3 IMPLEMENTATION DETAILS

The main baseline in our experiments is the LA loss (Menon et al., 2021) with Adapter+ tuning (Steitz & Roth, 2024). For this baseline, if not specified, we use a ViT-B/16 model (Dosovitskiy et al., 2021) pre-trained on ImageNet-21k (IN-21k) (Russakovsky et al., 2015) and train it with a SGD optimizer with learning rate $5 \times 10^{-4}$, weight decay $10^{-3}$, momentum 0.9, batch size 128, and image resolution $224 \times 224$. For all datasets except iNaturalist, training follows a cosine learning rate schedule with a linear warm-up over the first 10 epochs, for a total of 100 epochs. For iNaturalist, we adopt 2 warm-up epochs and train for 20 epochs in total, with a $10 \times$ larger learning rate. For all other baseline comparisons (except LA loss with Adapter+), the exact hyperparameters are provided in the released code scripts.

## D.4 REPRODUCIBILITY

`PTClf` is implemented in Pytorch and timm. Experiments are conducted on NVIDIA H20-96GB GPUs. To guarantee reproducibility, our full implementation will be publicly released; the anonymized code is available at https://anonymous.4open.science/r/PTClf.

## E MORE EXPERIMENTAL RESULTS

### E.1 ABLATION STUDIES

**Comprehensive ablation analysis of channel behavior.** Figs. xviii and xix illustrate the channel behavior after incorporating each component of our proposed `PTClf`. As shown in Fig. xviii, **PTInit** effectively transfers the upstream class-discriminative channels for downstream tail classes, leading to a denser distribution. Interestingly, **PTReg** also alleviates the imbalance by learning from the distribution of pre-trained class anchors. As shown in Fig. xix, **PTReg** exhibits channel behavior

more closely aligning with the oracle shown in Fig. 4. Note that **PTInit** also demonstrates improved behavior; however, its learned classifier remains less aligned with the oracle compared to **PTReg**, as highlighted in Fig. xi. We attribute this gap to the suboptimal initial label mapping (refer to Fig xiv), whereas **PTReg** can better exploit the well-mapped upstream classes during fine-tuning, leading to closer alignment with the oracle.

**Generalizability across architectures.** To investigate the architectural generalizability of `PTClf`, we conduct evaluations on a broad set of model families:

- **ViT & Swin.** We assess `PTClf` on larger-scale ViT variants as well as Swin Transformers.

- **CLIP.** Although CLIP does not include an explicit pre-trained classifier, our approach can still be naturally extended to it. The central idea of `PTClf` is to reuse the upstream knowledge encoded in the pre-trained model for data-scarce classification, rather than relying solely on the classifier itself. Specifically, we regard CLIP's text encoder as a classifier enriched with strong linguistic priors. To leverage these priors, we apply the IN-21k class names[3] to construct a set of text embeddings as pre-trained class anchors, which are then used to guide the fine-tuning of the text encoder (viewed as a pre-trained classifier).

Table v: **Performance across diverse architectures.**

| Model | C100-IR200 | Places-LT | Aircraft-LT |
|---|---|---|---|
| ViT-B/16 | 85.2 | 45.9 | 30.7 |
| + PTClf | **88.1** | **47.9** | **32.7** |
| ViT-L/16 | 87.4 | 47.2 | 40.6 |
| + PTClf | **88.6** | **48.0** | **41.2** |
| ViT-H/14 | 88.6 | 47.9 | 44.3 |
| + PTClf | **89.5** | **48.6** | **45.1** |
| Swin-B | 83.5 | 48.1 | 36.7 |
| + PTClf | **84.9** | **49.2** | **37.7** |
| CLIP-ViT-B/16 | 77.2 | 51.0 | 42.3 |
| + PTClf | **78.2** | **51.6** | **43.4** |
| DINOv2-B | 86.1 | 47.2 | 51.4 |
| + PTClf | **87.2** | **48.0** | **52.1** |

- **Self-Supervised Models.** Purely self-supervised models trained without a pre-trained classifier pose a distinct challenge for `PTClf`. Nevertheless, many popular self-supervised models provide ImageNet linear-probe checkpoints[4][5][6][7] that do include a classifier, which broadens the practical applicability of our method to those settings.

As shown in Tab. v, `PTClf` consistently improves performance across these diverse architectures, demonstrating superior architectural generalizability.

**Ablation analysis of label mapping strategies.** Naïve label mapping may not be the most effective design choice. Consequently, we further introduce three alternatives:

- **Semantic Matching.** We align downstream and upstream classes using label semantic similarity scores computed by a LLM.

- **Anchor Similarity.** Since upstream samples or embeddings are not available during downstream fine-tuning, we instead align labels based on the similarity between class anchors of the re-trained and pre-trained classifiers.

- **Probability-based Mixed Mapping.** Rather than assigning only a single pre-trained class anchor to each downstream class, we construct new mixed anchors for the re-trained classifier as: $\mathbf{w}'_{k^{\mathrm{r}}} = \sum_{k^{\mathrm{p}} \in [K^{\mathrm{p}}]} p_{k^{\mathrm{p}}(k^{\mathrm{r}})} \mathbf{w}_{k^{\mathrm{p}}}$, where $p_{k^{\mathrm{p}}(k^{\mathrm{r}})}$ denotes the normalized prediction frequency of upstream class $k^{\mathrm{p}}$ for downstream class $k^{\mathrm{r}}$.

As shown in Tab. vi, both semantic matching and probability-based mixed mapping slightly outperform the naïve frequency-based strategy. Nevertheless, our goal is to design a simple and effective approach, rather than rely on a more sophisticated alignment pipeline. For this reason, we keep adopting the frequency-based mapping as our default design choice.

Table vi: **Performance under different mapping strategies.**

| Method | C100-IR200 | Places-LT | Aircraft-LT |
|---|---|---|---|
| Baseline | 85.2 | 45.9 | 30.7 |
| Our Mapping | 88.1 | 47.9 | 32.7 |
| Semantic Matching | 87.6 | 48.4 | **32.9** |
| Anchor Similarity | 82.6 | 44.9 | 30.2 |
| Mixed Mapping | **88.7** | **48.5** | 31.8 |

## E.2    BENCHMARK RESULTS

**Benchmark Results across LTL Strategies.** Fig. vii presents detailed results on the FGVC-LT benchmark. As shown, our proposed `PTClf` achieves a clear performance gain on FGVC-LT, demonstrating superior effectiveness in fine-grained downstream adaptation.

---

[3]For simplicity, we directly use the IN-21k class names without additional curation.

[4]https://huggingface.co/facebook/dinov2-base-imagenet1k-1-layer

[5]https://huggingface.co/timm/hiera_base_plus_224.mae_in1k_ft_in1k

[6]https://huggingface.co/BobMcDear/vit_base_patch16_mae_in1k_224

[7]https://huggingface.co/lightly-ai/simclrv1-imagenet1k-resnet50-1x

Table vii: **Benchmark Results across various LTL strategies on FGVC-LT.** Results are presented in % using ViT-B/16 pre-trained on IN-21k. The baseline here is Adapter+ (Steitz & Roth, 2024) with different LTL strategies. [†‡]Same as Tab. 2. [*]Flower has no 'Median' classes, as its largest class contains only 10 samples.

| Method | Aircraft-LT | | | CUB-LT | | | Flowers-LT | | | Cars-LT | | | Dogs-LT | | | Average | | |
|---|---|---|---|---|---|---|---|---|---|---|---|---|---|---|---|---|---|---|
| | A | M$^†$ | T | A | M$^†$ | T | A | M$^{†*}$ | T | A | M$^†$ | T | A | M$^†$ | T | A | M$^†$ | T |
| CE | 28.5 | 66.4 | 11.7 | 62.9 | **90.0** | 59.2 | 81.6 | - | 81.6 | 37.2 | 74.0 | 17.8 | 63.9 | 87.9 | 48.0 | 54.8 | 79.6 | 43.7 |
| + `PTClf` | **30.2** | **68.0** | **13.4** | **68.7** | 89.6 | **65.9** | **92.7** | - | **92.7** | **38.6** | **75.7** | **19.0** | **87.3** | **93.3** | **83.2** | **63.5** | **81.6** | **54.8** |
| DODA (Wang et al., 2024) | 35.7 | 63.8 | 23.2 | 69.1 | 88.8 | 66.5 | 90.5 | - | 90.5 | 45.2 | 70.3 | 32.0 | 74.2 | 86.8 | 65.8 | 62.9 | 77.4 | 55.6 |
| + `PTClf` | **37.3** | **65.6** | **24.7** | **71.8** | **89.5** | **69.4** | **94.5** | - | **94.5** | **46.6** | **71.3** | **33.6** | **89.5** | **92.2** | **87.6** | **67.9** | **79.7** | **62.0** |
| cRT (Kang et al., 2020) | 40.1 | 68.6 | 27.5 | 67.2 | 87.0 | 64.6 | 95.9 | - | 95.9 | 52.3 | 75.2 | 40.2 | 70.8 | 86.9 | 60.2 | 65.3 | 79.4 | 57.7 |
| + `PTClf` | **40.3** | **68.9** | **27.6** | **70.8** | **88.0** | **68.5** | **96.8** | - | **96.8** | **52.8** | **75.4** | **40.9** | **89.0** | **90.6** | **87.9** | **69.9** | **80.7** | **64.3** |
| LOS (Sun et al., 2025) | 10.6 | **26.2** | 3.7 | 20.3 | **49.6** | 16.4 | 11.8 | - | 11.8 | 18.2 | **43.4** | 5.0 | 32.0 | 63.0 | 11.5 | 18.6 | **45.6** | 9.7 |
| + `PTClf` | **14.7** | 19.0 | **12.9** | **34.5** | 13.5 | **37.3** | **20.5** | - | **20.5** | **18.3** | 22.9 | **15.9** | **77.4** | **64.0** | **86.2** | **33.1** | 29.8 | **34.5** |
| LA (Menon et al., 2021) | 30.7 | 57.6 | 18.9 | 65.3 | 87.5 | 62.3 | 84.4 | - | 84.4 | 39.6 | 65.0 | 26.2 | 70.9 | 86.0 | 60.8 | 58.2 | 74.0 | 50.5 |
| + `PTClf` | **32.7** | **58.8** | **21.1** | **69.4** | **88.7** | **66.8** | **93.5** | - | **93.5** | **41.0** | **66.1** | **27.7** | **89.2** | **92.5** | **87.0** | **65.1** | **76.6** | **59.2** |
| PaCo (Cui et al., 2021) | 32.0 | **47.5** | 25.1 | 73.7 | 85.9 | 72.0 | 95.9 | - | 95.9 | 45.2 | **58.3** | 38.3 | 78.9 | 82.2 | 76.7 | 65.1 | 68.5 | 61.6 |
| + `PTClf` | **32.3** | 44.4 | **26.8** | **75.9** | **86.9** | **75.6** | 95.9 | - | 95.9 | **45.3** | 57.8 | **38.7** | **87.2** | **87.2** | **87.1** | **67.3** | **69.1** | **64.8** |
| ResLT (Cui et al., 2022) | 39.4 | **66.1** | 27.7 | 74.8 | 88.5 | 73.0 | 96.2 | - | 96.2 | 51.6 | 73.3 | 40.1 | 77.7 | 88.3 | 70.7 | 67.9 | 79.0 | 61.5 |
| + `PTClf` | **40.1** | 65.2 | **28.9** | **76.4** | **89.5** | **74.6** | **96.5** | - | **96.5** | **52.1** | **73.5** | 40.8 | **88.6** | **91.6** | **86.6** | **70.7** | **80.0** | **65.5** |
| LPT (Dong et al., 2023)$^‡$ | - | - | - | - | - | - | - | - | - | - | - | - | - | - | - | - | - | - |
| `PTClf`$^‡$ | **43.7** | **72.4** | **31.0** | **78.1** | **89.8** | **76.6** | **97.6** | - | **97.6** | **59.6** | **81.9** | **47.9** | **88.9** | **90.3** | **88.0** | **73.6** | **83.6** | **68.2** |

**A**: All; **M**: Median; **T**: Tail; **Aircraft**: FGVCAircraft; **CUB**: CUB-200-2011; **Flowers**: Oxford Flowers; **Cars**: Stanford Cars; **Dogs**: Stanford Dogs.

Table viii: **Benchmark Results across various PEFT approaches.** Results are presented in % using ViT-B/16 pre-trained on IN-21k. The baseline here is the LA loss (Menon et al., 2021) within different PEFT approaches. [†]Same as Tab. 2.

| Method | C100-IR200 | | | C100-IR100 | | | C100-IR50 | | | Places-LT | | | iNaturalist | | | FGVC-LT | | |
|---|---|---|---|---|---|---|---|---|---|---|---|---|---|---|---|---|---|---|
| | A | H | T | A | H | T | A | H | T | A | H | T | A | H | T | A | M$^†$ | T |
| Reparameter Tuning | 85.0 | 92.2 | 77.3 | 87.7 | 92.1 | 83.2 | 89.8 | 92.2 | 88.6 | 45.7 | 48.1 | 41.4 | 78.1 | 72.3 | 79.0 | 58.4 | 74.1 | 50.7 |
| + `PTClf` | **87.9** | **92.7** | **83.5** | **89.9** | **92.4** | **87.4** | **90.6** | **92.3** | **90.1** | **47.6** | **48.9** | **44.9** | **78.7** | **74.2** | **79.3** | **64.8** | **76.2** | **58.9** |
| Prompt Tuning | 85.3 | 92.4 | 77.6 | 87.8 | 92.0 | 84.5 | 90.0 | 91.8 | 89.0 | 45.8 | 47.5 | 41.8 | 77.1 | 71.6 | 78.8 | 57.8 | 73.7 | 49.9 |
| + `PTClf` | **88.2** | **92.9** | **84.5** | **89.8** | **92.1** | **87.4** | **90.5** | **92.0** | **90.8** | **47.6** | **47.8** | **45.3** | **78.0** | **73.2** | **79.0** | **64.1** | **75.4** | **58.5** |
| Adapter Tuning | 85.2 | 92.4 | 77.4 | 87.9 | 92.3 | 83.2 | 89.8 | 92.1 | 88.8 | 45.9 | 48.1 | 41.6 | 78.6 | 73.0 | 79.4 | 58.2 | 74.0 | 50.5 |
| + `PTClf` | **88.1** | **92.8** | **84.1** | **89.8** | **92.4** | **87.3** | **90.6** | **92.2** | **90.4** | **47.9** | **48.6** | **45.8** | **79.4** | **74.9** | **79.7** | **65.1** | **76.6** | **59.2** |

**A**: All; **H**: Head; **M**: Median; **T**: Tail; **C100**: CIFAR100.

**Benchmark Results across PEFT Methods.** Our proposed `PTClf` paradigm is broadly applicable to different fine-tuning methods. To investigate this, we conduct evaluations on the long-tailed benchmarks across three representative PEFT approaches, w/ or w/o `PTClf`, including: reparameter tuning LoRA (Hu et al., 2022), prompt tuning VPT (Jia et al., 2022), and adapter tuning Adapter+ (Steitz & Roth, 2024). As shown in Tab. viii, our proposed `PTClf` method enhances performance across all PEFT approaches, with particularly notable gains on tail classes. Interestingly, different PEFT methods yield similar results when the hyperparameters are appropriately chosen, consistent with the findings of Mai et al. (2025), suggesting that the choice of PEFT methods has a negligible effect.

# F LIMITATION AND FUTURE WORK

Despite demonstrating promising performance on various downstream tasks, our proposed `PTClf` paradigm still encounters a limitation. In most scenarios, the number of upstream classes exceeds that of downstream classes; however, exceptions exist. For instance, the iNaturalist dataset (Van Horn et al., 2018) contains 8k+ downstream classes, surpassing the 1k upstream classes in IN-1k pre-training. In such cases, `PTClf` cannot be directly applied. This highlights an important direction for future work: how can we leverage a limited set of upstream class anchors to effectively support a larger set of downstream classes?

Nevertheless, we argue that this limitation may become less critical as pre-training continues to evolve. In the era of web-scale data, pre-training data are expanding rapidly and are likely to encompass the majority of downstream tasks. Consequently, the utility of the pre-trained classifier will

continue to grow. We therefore believe that the `PTClf` paradigm developed here can inspire future approaches to downstream adaptation.

## G LLM USAGE STATEMENT

In this work, large language models (LLMs) were used solely for text refinement. No original content, methodology, or ideas were generated by LLMs.

