# OpenReview forum: "Beyond Re-Training from Scratch: Exploiting the Pre-Trained Classifier for Long-Tailed Learning"
_ICLR.cc/2026/Conference — Submitted to ICLR 2026_

### Official Review · Reviewer_EgUh · 2025-10-30

**Soundness:** 2
**Presentation:** 3
**Contribution:** 2
**Rating:** 4
**Confidence:** 4

**Summary:**

This paper proposes a method for long-tail classification that aims to leverage a pre-trained foundation classification model to improve performance on tail classes. Authors claim that approaches using foundation models only rely on the feature representation, but fail to leverage the classifier. They provide an analysis of long tailed classifiers, showing poor representation power, and show that better results can be achieved by transferring knowledge from a foundation pre-trained classifier to similar target classes, by using the foundation classifier as initialisation. Experiments show that this approach can be used in a plug and play manner and boost performance of pre-existing methods.

**Strengths:**

The paper is clearly written and thoroughly analyses the problem studied, highlighting the challenges of long tailed learning. While certain conclusions are not surprising (long tail classification fails on tail classes due to lack of data), authors clearly discuss their hypothesis and highlight why classifiers fail. The thoroughness and attention to detail is appreciated in this work.

Similarly, the experimental evaluation is thorough, providing multiple ablations and evaluating the approach from multiple angles. I particularly appreciated the experiment with out of distribution data, which was an important scenario to test and validate.

The method is simple, can be used in a plug and play manner, and seems to consistently boost performance.

**Weaknesses:**

I see two main weaknesses with this work:

1-	It is quite impractical, as it limits to foundation models pre-trained for classification tasks, learning a classification module. This means that very popular foundation models, known for providing high performance boosts, that do not have a classification head are not usable. This includes models like CLIP, or self supervised models trained at scale. This substantially reduces the applicability of the method as foundation models rarely have a classification head available.

2-	A second weakness is the main hypothesis of this work, which is that pre-trained classifiers can be used to boost performance of rare classes by transferring related knowledge. While this is a sensible assumption, this is not fully demonstrated by experiments in this paper. Authors highlight that classifier channels trained on rare classes have lower magnitude and non discriminative feature channels. It is possible that performance gains are not due to knowledge transfer, but to a better alignment between feature and classifier at initialisation time, as well as a better separation between classifier channels. If knowledge transfer was the key reason behind the performance gain, one could expect results on OOD data to be worsened by transferring irrelevant knowledge to tail classes. A valuable experiment to verify the hypothesis would be replacing label mapping with a random mapping.

Lastly, the paper is quite crowded, which means that certain experiments lack relevant detail or clarity. While it is great to see a large set of experiments, it is important to provide sufficient details to allow the reader to get valuable information out of it. It would be better to focus on key experiments in the main paper, and move some (e.g. low shot experiments) in the appendix.

**Questions:**

-	Overall, this is an interesting paper, aiming to understand how long tail classifiers learn. My main concern is the unpractical requirement to have a pre-trained classifier associated with strong foundation backbones used for the task.
-	It would be great to verify the knowledge transfer hypothesis to understand more clearly the importance of semantic mapping vs a better distribution and alignment with the pre-trained backbone.

---

> ### Author Response · Authors · 2025-11-23
> **Rebuttal by Authors [1/2]**
>
> Thank you for the thoughtful and detailed evaluation of our manuscript. We address your concerns and questions below:
>
>
> > **[W1/Q1].** It is quite impractical, as it limits to foundation models pre-trained for classification tasks, learning a classification module. This means that very popular foundation models, known for providing high performance boosts, that do not have a classification head are not usable. This includes models like CLIP, or self supervised models trained at scale. This substantially reduces the applicability of the method as foundation models rarely have a classification head available. / Overall, this is an interesting paper, aiming to understand how long tail classifiers learn. My main concern is the unpractical requirement to have a pre-trained classifier associated with strong foundation backbones used for the task.
>
> Thank you for the insightful comment. To address your concern about practicality, we extend our method to broader foundation models, including CLIP and self-supervised models.
>
> - **PTClf can be effectively extended to open-vocabulary models like CLIP.** Indeed, PTClf was initially designed for models with a pre-trained classification head. More broadly, however, **the core idea behind PTClf is to reuse the abundant knowledge within the pre-trained model for knowledge-scarce classification**, rather than relying strictly on the classifier itself. **This idea can be naturally applied to models like CLIP.** In specific, we regard the text encoder in CLIP as a classification module enriched with strong linguistic priors. To reuse these priors, we apply the IN-21k class names* to construct a set of text embeddings as pre-trained class anchors, which are then used to guide the fine-tuning of the text encoder (treated as a pre-trained classifier). The results are presented in the following table:
>
>   |         | C100-IR200          | Places-LT           | FGVCAircraft-LT     |
>   | ------- | ------------------- | ------------------- | ------------------- |
>   | CLIP    | 77.2                | 51.0                | 42.3                |
>   | + PTClf | **78.2** (**+1.0**) | **51.6** (**+0.6**) | **43.4** (**+1.1**) |
>
>   As shown, PTClf brings consistent performance gains on CLIP.
>
> - **PTClf remains effective for self-supervised models with an available linear-probe classifier.** We acknowledge that purely self-supervised models trained without any classification head pose a distinct challenge for PTClf. However, **many popular self-supervised models provide ImageNet linear-probe checkpoints that include a classifier [1-4], which makes our method applicable in practice.** To verify this, we conduct experiments on DINOv2 [1], with results shown below:
>
>   |         | C100-IR200          | Places-LT           | FGVCAircraft-LT     |
>   | ------- | ------------------- | ------------------- | ------------------- |
>   | DINOv2  | 86.1                | 47.2                | 51.4                |
>   | + PTClf | **87.2** (**+1.1**) | **48.0** (**+0.8**) | **52.1** (**+0.7**) |
>
>   As shown, PTClf also brings consistent improvements to self-supervised models. Nonetheless, we acknowledge that extending PTClf to completely classifier-free self-supervised models remains a meaningful and non-trivial direction, which we plan to explore in our future work.
>
> *For simplicity, we directly use the IN-21k class names without additional curation.
>
> [1] https://huggingface.co/facebook/dinov2-base-imagenet1k-1-layer.
>
> [2] https://huggingface.co/timm/hiera_base_plus_224.mae_in1k_ft_in1k.
>
> [3] https://huggingface.co/BobMcDear/vit_base_patch16_mae_in1k_224.
>
> [4] https://huggingface.co/lightly-ai/simclrv1-imagenet1k-resnet50-1x.

---

> ### Author Response · Authors · 2025-11-23
> **Rebuttal by Authors [2/2]**
>
> > **[W2/Q2].** A second weakness is the main hypothesis of this work, which is that pre-trained classifiers can be used to boost performance of rare classes by transferring related knowledge. While this is a sensible assumption, this is not fully demonstrated by experiments in this paper. Authors highlight that classifier channels trained on rare classes have lower magnitude and non discriminative feature channels. It is possible that performance gains are not due to knowledge transfer, but to a better alignment between feature and classifier at initialisation time, as well as a better separation between classifier channels. If knowledge transfer was the key reason behind the performance gain, one could expect results on OOD data to be worsened by transferring irrelevant knowledge to tail classes. A valuable experiment to verify the hypothesis would be replacing label mapping with a random mapping. / It would be great to verify the knowledge transfer hypothesis to understand more clearly the importance of semantic mapping vs a better distribution and alignment with the pre-trained backbone.
>
> **PTClf's gains mainly come from transferring relevant upstream knowledge to downstream classes, rather than from better channel distribution or alignment with the backbone.** We validate this from two perspectives:
>
> - **Random (uncorrelated) mapping hurts performance, indicating that distribution or alignment is not the main factor.** Following the reviewer's suggestion, we replace the original label mapping with a random mapping. Specifically, random mapping retains the potential benefits from improved distribution or alignment offered by pre-trained classifiers, but explicitly removes label correlations. As shown in the table below, performance consistently declines under random mapping, thereby confirming that the observed gains are not simply explained by distribution or alignment effects.
>
>   |                       | C100-IR200          | Places-LT           | FGVCAircraft-LT     |
>   | --------------------- | ------------------- | ------------------- | ------------------- |
>   | Baseline              | 85.2                | 45.9                | 30.7                |
>   | Label Mapping (paper) | **88.1** (**+2.9**) | **47.9** (**+2.0**) | **32.7** (**+2.0**) |
>   | Random Mapping        | 84.6 (-0.6)         | 45.8 (-0.1)         | 30.4 (-0.3)         |
>
> - **Gains scale with upstream-downstream knowledge correlation, supporting knowledge transfer as the driving factor.** As pointed out by the reviewer, an intuitive way to verify whether knowledge transfer is the main factor is to examine how gains vary with the knowledge relatedness. If knowledge transfer is indeed central, PTClf should be less effective on weakly related (OOD) tasks and more effective on closely related (ID) tasks. Motivated by this, we visualize knowledge correlation versus performance gain across datasets ranging from extreme OOD to ID:
>
>   1. *Extreme OOD* (EuroSAT, satellite observation):
>
>      Correlation: https://drive.google.com/file/d/1zHYaog_QDi5eUAS1tspEtwJJyD_p3qb5 — Gain: 0.4%
>   2. *Weak OOD* (DTD, textures):
>
>      Correlation: https://drive.google.com/file/d/1GdJdQ5T_eD_28UT-Os6oRZMmgNYOK9oU — Gain: 0.7%
>   3. *Moderate overlap* (CIFAR100, natural images):
>
>      Correlation: https://drive.google.com/file/d/1dHwdLmDJaKVigm-1NReW4kzud7kPue7u — Gain: 2.9%
>   4. *Complete overlap* (Stanford Dogs, subset of ImageNet):
>
>      Correlation: https://drive.google.com/file/d/1f4TmDaIkQ8jfB1n_ePO5eiRzRgr3_YLx — Gain: 18.3%
>
>   The trend is clear: the more semantically aligned the upstream-downstream mapping, the larger the PTClf improvement. **Even for OOD datasets like EuroSAT and DTD, mapped upstream labels retain weak but non-zero knowledge correlations (e.g., similar visual attributes), thereby exhibiting small but measurable gains.** For ID datasets like CIFAR100 and Stanford Dogs, correlation is much higher and the resulting gains are substantially larger. Such a clear scaling effectively supports that knowledge transfer is indeed the main driver behind PTClf's gains.

---

### Official Review · Reviewer_6hC2 · 2025-10-30

**Soundness:** 2
**Presentation:** 3
**Contribution:** 2
**Rating:** 4
**Confidence:** 3

**Summary:**

This paper proposes **PTClf**, a fine-tuning paradigm for long-tailed recognition that reuses the *pre-trained classifier* rather than discarding it.  The method aligns downstream classes to the most similar upstream classes using prediction frequency, initializes the new classifier with the corresponding pre-trained weights, and regularizes it to stay close during training.
The authors claim that this approach improves long-tail performance and even generalizes to OOD and DG (domain generalization) settings.  Empirically, the paper reports consistent gains (up to +10 pp) on CIFAR100-LT, IN-LT, and smaller improvements on SVHN and DTD.

**Strengths:**

- **Clear and reproducible formulation:** The proposed PTClf framework is simple, easy to implement, and described clearly with sufficient ablation details.
- **Strong empirical results:** Consistent performance gains across multiple long-tail benchmarks (CIFAR100-LT, IN-LT, iNat-LT) and compatibility with existing rebalancing methods (cRT, LA, PaCo).
- **Thorough ablations:** The paper includes solid analyses on mapping, initialization, and regularization effects, showing internal consistency within its setup.

**Weaknesses:**

- **Dependence on pretraining proximity and potential semantic overlap**.
The performance gains seem to depend on how closely the downstream dataset aligns with the pretraining distribution.
Most of the evaluated datasets (CIFAR100-LT, IN-LT, iNaturalist, FGVC-LT) share visual or semantic characteristics with ImageNet-21K.
Because PTClf reuses the pre-trained classifier and aligns classes based on prediction similarity, it may inadvertently benefit from overlapping or conceptually related categories. This is not necessarily problematic, but it would be helpful for the authors to clarify whether any precautions were taken to avoid **category leakage** and how much of the gain stems from this overlap versus genuine transferability.


- **OOD and DG settings may not represent real-world long-tailed challenges**
The datasets labeled as OOD or DG (SVHN, DTD, CIFAR10-DG, IN-DG) largely remain within the *natural image* domain and differ mainly by appearance, texture, or augmentation rather than by substantial semantic shift.  As such, the demonstrated generalization might be more modest than the “OOD/DG” terminology implies.  In practical scenarios, long-tailed problems often arise in **naturally OOD settings**—for example, medical imaging, satellite observations, or rare biological species—where data scarcity and domain shift coincide.  Evaluating the method on such data could provide stronger evidence of its real-world robustness.

- **Limited comparison breadth in OOD/DG evaluations**
For OOD and DG experiments, PTClf is compared mainly with *standard fine-tuning*, without including other competitive baselines such as LDAM-DRW, cRT, LWS, LADE, MixStyle, SWAD, or Fishr.  Including these methods could better contextualize PTClf’s advantages and help clarify whether the reported gains are specific to classifier reuse or simply a reflection of a weaker baseline.  This is not a major flaw but an area that could make the empirical evidence more convincing.



- **Improvements appear correlated with similarity and imbalance severity**
Across datasets, PTClf’s gains increase when the downstream task is more similar to ImageNet or when the imbalance ratio is extreme (e.g., D-IR40, C100-IR200). This trend suggests that the method’s strength may lie in **leveraging pretraining priors** when limited data are available rather than learning representations that generalize to unseen domains.
While such prior reuse is practically valuable, it would be helpful if the authors discussed this trade-off more explicitly.


- **Conceptual overlap with open-vocabulary or zero-shot approaches**
If PTClf primarily serves to inject a classifier prior learned from large-scale pretraining, one might ask how it compares to open-vocabulary methods such as CLIP or SigLIP.  These models achieve a similar goal—leveraging pretrained semantic priors—through language-aligned embeddings and often handle OOD categories more gracefully.  A brief discussion of this connection could clarify PTClf’s niche and its relevance in the evolving landscape of multimodal pretraining.

**Questions:**

1. How does PTClf perform when pre-training and downstream label spaces are disjoint (e.g., non-overlapping ImageNet splits)?
2. Can you quantify or visualize the correlation between dataset similarity (to ImageNet) and the observed gains?
3. Why were stronger baselines (SWAD, MixStyle, LADE) omitted in OOD/DG comparisons?
4. Could PTClf still offer value when using an open-vocabulary model such as CLIP, or does it become redundant?

---

> ### Author Response · Authors · 2025-11-23
> **Rebuttal by Authors [1/3]**
>
> Thank you for your thoughtful evaluation and valuable feedback. We address your concerns and questions below:
>
>
> > **[W1].** Dependence on pretraining proximity and potential semantic overlap. The performance gains seem to depend on how closely the downstream dataset aligns with the pretraining distribution. Most of the evaluated datasets (CIFAR100-LT, IN-LT, iNaturalist, FGVC-LT) share visual or semantic characteristics with ImageNet-21K. Because PTClf reuses the pre-trained classifier and aligns classes based on prediction similarity, it may inadvertently benefit from overlapping or conceptually related categories. This is not necessarily problematic, but it would be helpful for the authors to clarify whether any precautions were taken to avoid category leakage and how much of the gain stems from this overlap versus genuine transferability.
>
> **The improvements of PTClf stem from both overlapping classes and knowledge transfer from non-overlapping classes.** To investigate how PTClf's gains depend on class relatedness, we perform an additional analysis using WordNet path similarity to quantify the relatedness of each upstream-downstream label pair. We specifically divide label pairs into three groups:
>
> - *Overlap*: identical labels or hypernym-hyponym pairs (path similarity $\in$ [0.9, 1.0]);
> - *Similar*: semantically related but not hierarchically identical categories (path similarity $\in$ [0.6, 0.9));
> - *Distant*: weakly related or unrelated categories (path similarity $\in$ [0.0, 0.6)).
>
> The results are presented at https://drive.google.com/file/d/12You0BD4tBpLVg9QkdsiIZQWY_KC_fDs. As shown in the figure, PTClf delivers notable improvements across *all* three groups, i.e., Overlap, Similar, and Distant. More importantly, the largest gains come from the Similar group rather than the Overlap group, while the Distinct group also exhibits clear and comparable improvements. This demonstrates that **PTClf benefits not only from overlapping classes but also from genuine knowledge transfer among non-overlapping classes.**
>
> Additionally, we would like to emphasize that **reusing pre-existing knowledge is precisely the goal of PTClf.** Since the re-trained classifier struggles with data-scarce classes, PTClf aims to exploits the knowledge already encoded in the pre-trained model, *including both overlapping and non-overlapping*, to provide effective guidance.
>
> &nbsp;
>
>
>
> > **[W2/Q1].** OOD and DG settings may not represent real-world long-tailed challenges. The datasets labeled as OOD or DG (SVHN, DTD, CIFAR10-DG, IN-DG) largely remain within the *natural image* domain and differ mainly by appearance, texture, or augmentation rather than by substantial semantic shift. As such, the demonstrated generalization might be more modest than the “OOD/DG” terminology implies. In practical scenarios, long-tailed problems often arise in naturally OOD settings—for example, medical imaging, satellite observations, or rare biological species—where data scarcity and domain shift coincide. Evaluating the method on such data could provide stronger evidence of its real-world robustness. / How does PTClf perform when pre-training and downstream label spaces are disjoint (e.g., non-overlapping ImageNet splits)?
>
> We clarify that SVHN and DTD do not belong to the natural image domain, as they comprise digit-only and texture-only images, respectively. Nonetheless, these datasets can indeed be regarded as moderately OOD w.r.t. ImageNet, rather than extremely OOD. Following your suggestion, **we further evaluate our method on three more extreme OOD long-tailed datasets: ISIC-LT (medical imaging), EuroSAT-LT (satellite observation), and RareSpecies-LT (rare biological species).** The results are shown in the following table:
>
> |          | ISIC-LT     | EuroSAT-LT          | RareSpecies-LT      |
> | -------- | ----------- | ------------------- | ------------------- |
> | Baseline | **69.8**    | 89.5                | 53.6                |
> | + PTClf  | 69.7 (-0.1) | **89.9** (**+0.4**) | **54.4** (**+0.8**) |
>
> As shown, PTClf yields modest gains on EuroSAT-LT and RareSpecies-LT, but leads to a marginal decrease on ISIC-LT. Overall, the gains are much less pronounced in these extreme OOD scenarios. We attribute this to the fact that the foundation model itself is less effective on domains that deviate significantly from ImageNet \[1][2]; consequently, the utility of its pre-trained classifier is also naturally reduced on such data.
>
> [1] Ju, Lie, et al. MONICA: Benchmarking on Long-tailed Medical Image Classification. ArXiv'24.
>
> [2] Raghu, Maithra, et al. Transfusion: Understanding transfer learning for medical imaging. NeurIPS'19.

---

> ### Author Response · Authors · 2025-11-23
> **Rebuttal by Authors [2/3]**
>
> > **[W3/Q3].** Limited comparison breadth in OOD/DG evaluations. For OOD and DG experiments, PTClf is compared mainly with *standard fine-tuning*, without including other competitive baselines such as LDAM-DRW, cRT, LWS, LADE, MixStyle, SWAD, or Fishr. Including these methods could better contextualize PTClf’s advantages and help clarify whether the reported gains are specific to classifier reuse or simply a reflection of a weaker baseline. This is not a major flaw but an area that could make the empirical evidence more convincing. / Why were stronger baselines (SWAD, MixStyle, LADE) omitted in OOD/DG comparisons?
>
> **PTClf can be added to different OOD/DG baselines and consistently boosts performance.** To address your concern, we conduct additional experiments with a classic OOD/DG method, SWAD [1], and a more advanced method, SPD [2], as shown in the following table. Since all OOD/DG evaluations (including the table below and Tables 4 and ii in the main paper) are already performed with the advanced balanced LA loss [3], we do not include comparisons with other long-tailed strategies such as LADE or cRT.
>
> |          | C10-LT              | C10-DG              | IN-LT               | IN-DG               |
> | -------- | ------------------- | ------------------- | ------------------- | ------------------- |
> | Baseline | 95.3                | 91.6                | 79.6                | 39.9                |
> | + PTClf  | **96.3** (**+1.0**) | **94.0** (**+2.4**) | **82.7** (**+3.1**) | **43.6** (**+3.7**) |
> | SWAD [1] | 91.8                | 86.8                | 80.1                | 40.8                |
> | + PTClf  | **95.7** (**+3.9**) | **91.1** (**+4.3**) | **82.9** (**+2.8**) | **44.1** (**+3.3**) |
> | SPD [2]  | 95.2                | 94.7                | 80.4                | 42.9                |
> | + PTClf  | **97.7** (**+2.5**) | **97.6** (**+2.9**) | **83.4** (**+2.0**) | **44.8** (**+1.9**) |
>
> As shown, PTClf consistently improves performance across these stronger baselines, further demonstrating its plug-and-play effectiveness.
>
> [1] Cha, Junbum, et al. Swad: Domain generalization by seeking flat minima. NeurIPS'21.
>
> [2] Tian, Junjiao, Chengyue Huang, and Zsolt Kira. Rethinking weight decay for robust fine-tuning of foundation models. NeurIPS'24.
>
> [3] Menon, Aditya Krishna, et al. Long-tail learning via logit adjustment. ICLR'21.
>
> &nbsp;
>
> > **[W4].** Improvements appear correlated with similarity and imbalance severity. Across datasets, PTClf’s gains increase when the downstream task is more similar to ImageNet or when the imbalance ratio is extreme (e.g., D-IR40, C100-IR200). This trend suggests that the method’s strength may lie in leveraging pretraining priors when limited data are available rather than learning representations that generalize to unseen domains. While such prior reuse is practically valuable, it would be helpful if the authors discussed this trade-off more explicitly.
>
> Thank you for this valuable comment. We agree that PTClf's gains tend to be larger when the downstream task is more ImageNet-like or exhibits more severe imbalance. This behavior is in line with our design goal: **to reuse the class priors learned from ImageNet to support data-scarce tail classes.** When the downstream task is closer to ImageNet, these priors are more directly applicable; when tail classes have very few samples, leveraging such priors is often more effective than attempting to learn a reliable classifier from scarce data.
>
> We also acknowledge the trade-off: PTClf is less beneficial for *extremely* OOD (i.e., unseen) domains. For example, on the ISIC dataset (skin lesion imagery), we observe a slight performance drop (refer to W2/Q1). We attribute to the fact that the *entire* ImageNet-pretrained model is misaligned with the medical imaging domain \[1][2]. In other words, **when the upstream and downstream domains are largely orthogonal, reusing either its backbone or its classifier provides only marginal gains.** Nevertheless, **PTClf does show transferability to certain OOD domains.** As shown in our label mapping analysis on EuroSAT (https://drive.google.com/file/d/1zHYaog_QDi5eUAS1tspEtwJJyD_p3qb5), PTClf can exploit disjoint but semantically related ImageNet labels to guide classifier learning, indicating that PTClf can indeed generalize to some unseen domains.
>
> Overall, our aim is pragmatic: **for downstream tasks where ImageNet-pretrained models can be effectively adapted, PTClf is designed to make better use of upstream class knowledge, rather than to learn a classifier solely from imperfect data.**
>
> [1] Ju, Lie, et al. MONICA: Benchmarking on Long-tailed Medical Image Classification. ArXiv'24.
>
> [2] Raghu, Maithra, et al. Transfusion: Understanding transfer learning for medical imaging. NeurIPS'19.

---

> ### Author Response · Authors · 2025-11-23
> **Rebuttal by Authors [3/3]**
>
> > **[W5].** Conceptual overlap with open-vocabulary or zero-shot approaches. If PTClf primarily serves to inject a classifier prior learned from large-scale pretraining, one might ask how it compares to open-vocabulary methods such as CLIP or SigLIP. These models achieve a similar goal—leveraging pretrained semantic priors—through language-aligned embeddings and often handle OOD categories more gracefully. A brief discussion of this connection could clarify PTClf’s niche and its relevance in the evolving landscape of multimodal pretraining.
>
> This is an insightful question. Indeed, both PTClf and open-vocabulary models leverage the pre-trained priors of foundation models. However, **their goals are fundamentally different.** Open-vocabulary models such as CLIP introduce additional knowledge from other aspects (i.e., text), which can be viewed as **information augmentation**. In contrast, PTClf focuses on **information reuse**, i.e., directly reusing the knowledge already embedded in the pre-trained model itself *during downstream fine-tuning*. We regarded these ideas as complementary rather than competing. As discussed in Q4, our idea can also be applied to CLIP by reusing the priors from its text encoder, which leads to further performance improvements.
>
> &nbsp;
>
> > **[Q2].** Can you quantify or visualize the correlation between dataset similarity (to ImageNet) and the observed gains?
>
> **PTClf tends to deliver larger gains on datasets that are more closely aligned with ImageNet.** To address your question, we introduce the Fréchet Inception Distance (FID) to quantify the similarity between downstream tasks and upstream ImageNet, where lower scores indicate higher similarity. We visualize the performance gains w.r.t. FID at https://drive.google.com/file/d/19h4OWq8W-eUqeRJU45LjMSv4eOPzz6Qf. As shown in the figure, PTClf brings higher gains when the downstream data is more similar to ImageNet.
>
> &nbsp;
>
> > **[Q4].** Could PTClf still offer value when using an open-vocabulary model such as CLIP, or does it become redundant?
>
> **PTClf can indeed benefit open-vocabulary models like CLIP.** Specifically, we treat the text encoder in CLIP as a text-driven pre-trained classifier. Using the IN-21k class names*, we generate text embeddings as text-based class anchors and use them to guide fine-tuning of the text encoder. The results are shown in the table below:
>
> |         | C100-IR200          | Places-LT           | FGVCAircraft-LT     |
> | ------- | ------------------- | ------------------- | ------------------- |
> | CLIP    | 77.2                | 51.0                | 42.3                |
> | + PTClf | **78.2** (**+1.0**) | **51.6** (**+0.6**) | **43.4** (**+1.1**) |
>
> As demonstrated, PTClf also yields consistent performance improvements on CLIP.
>
> *For simplicity, we directly use the IN-21k class names without additional curation.

---

### Official Review · Reviewer_CDwk · 2025-10-31

**Soundness:** 2
**Presentation:** 3
**Contribution:** 2
**Rating:** 4
**Confidence:** 5

**Summary:**

This paper presents a comprehensive analysis of classifier behavior when fine-tuning pre-trained visual models on long-tailed downstream tasks. The authors show that the newly re-trained classifier often exhibits weakened discriminative ability and semantic awareness, with severe imbalances in class-discriminative channels and mislearning of general features for tail classes. They propose PTClf (Pre-Trained Classifier helps), a fine-tuning approach that initializes the classifier with pre-trained weights and adds a regularization term, PTReg. The paper includes comprehensive experiments supporting the effectiveness of the proposed method.

**Strengths:**

* The paper is well written and complete, with no obvious flaws.
* The analysis is solid and the methodology is well motivated. The figures are well designed and informative. The experiments provide sufficient evidence for the proposed method’s effectiveness.

**Weaknesses:**

* Overall novelty is limited. Leveraging the pre-trained classifier during fine-tuning dates back at least five years (see [1]).
* The analysis is conducted on specific classification pre-trained models and datasets. It is unclear whether the findings generalize to a broader range of pre-trained models, e.g., self-supervised models such as DINO or MAE.
* Results in Table ii lag behind those reported in [1], which used a ResNet-50 backbone five years ago (e.g., Stanford Cars vanilla fine-tuning reaches 87.20%, whereas the highest in Table iii is 59.6%). It is unclear whether evaluation protocols differ in the fine-grained fine-tuning setup. This uncertainty reduces confidence in the overall long-tailed visual fine-tuning benchmarks presented here.

[1] Co-Tuning for Transfer Learning. NeurIPS 2020.

**Questions:**

* The authors claim to focus on PEFT, long-tailed, visual tasks, but the methodology appears general. Can PTClf be extended to other domains, such as LLM fine-tuning?
* Can PTClf be applied to tasks beyond classification? The long-tailed problem is even more challenging in object detection.
* The assumption that the pre-trained model covers more classes usually does not hold (line 256). In your fine-grained benchmarks—e.g., Stanford Dogs, a subset of ImageNet with finer labels—the downstream task is more fine-grained than the pre-trained task. How do you ensure that the label mapping works well in this setup?

---

> ### Author Response · Authors · 2025-11-23
> **Rebuttal by Authors [1/3]**
>
> Thank you for your thorough reading and insightful feedback. We address your concerns and questions below:
>
>
> > **[W1].** Overall novelty is limited. Leveraging the pre-trained classifier during fine-tuning dates back at least five years (see [1]).
>
> Thank you for pointing this out. We would like to clarify our novelty from two perspectives:
>
> - **Our work focuses on the *classifier*, which is fundamentally different from Co-Tuning.** Specifically, Co-Tuning jointly leverages the pre-trained and re-trained classifiers to co-supervise the backbone, aiming to improve feature learning, while the re-trained classifier itself is still learned from scratch without guidance. PTClf instead uses the pre-trained classifier to assist the re-trained classifier, aiming to improve its classification capability. Consequently, the two approaches are in fact complementary. We also conduct experiments incorporating PTClf into Co-Tuning, as shown in the table below.
>
>   |                   | C100-IR200          | Places-LT           | FGVCAircraft-LT     |
>   | ----------------- | ------------------- | ------------------- | ------------------- |
>   | Baseline          | 85.2                | 45.9                | 30.7                |
>   | Co-Tuning [1]     | 86.1 (+0.9)         | 47.0 (+1.1)         | 31.6 (+0.9)         |
>   | PTClf             | 88.1 (+2.9)         | 47.9 (+2.0)         | 32.7 (+2.0)         |
>   | Co-Tuning + PTClf | **88.4** (**+3.2**) | **48.2** (**+2.3**) | **32.8** (**+2.1**) |
>
> - **The novelty of our work lies not only in the methodology, but more importantly in our analysis of classifiers in long-tailed learning.** Classifier bias has long been recognized as a key challenge in long-tailed learning, yet prior studies have mainly examined it at the norm level and focused solely on debiasing the skewed weight norms. By introducing channel-level metrics, we go one step deeper and uncover two previously overlooked issues—weakened discriminative ability and semantic awareness—and further reveal that both **primarily arise from sample scarcity in tail classes, rather than class imbalance itself**. This perspective is particularly insightful for long-tailed learning, as **existing works on classifier bias largely attribute it to the class imbalance**. We believe these findings offer meaningful insights for the field beyond the method itself.
>
> [1] You, Kaichao, et al. Co-tuning for transfer learning. NeurIPS 2020.

---

> ### Author Response · Authors · 2025-11-23
> **Rebuttal by Authors [2/3]**
>
> > **[W2].** The analysis is conducted on specific classification pre-trained models and datasets. It is unclear whether the findings generalize to a broader range of pre-trained models, e.g., self-supervised models such as DINO or MAE.
>
> **PTClf remains effective across a broad range of pre-trained models.** We conduct additional experiments on diverse pre-trained architectures, including ViT-L/16 and ViT-H/14 (ViT family), Swin-B (Swin family), CLIP-ViT-B/16 (CLIP), and DINOv2-B (DINO). The results are shown in the table below:
>
> | | C100-IR200          | Places-LT           | FGVCAircraft-LT     |
> | ---------------- | ------------------- | ------------------- | ------------------- |
> | ViT-B/16 (paper) | 85.2                | 45.9                | 30.7                |
> | + PTClf          | **88.1** (**+2.9**) | **47.9** (**+2.0**) | **32.7** (**+2.0**) |
> | ViT-L/16         | 87.4                | 47.2                | 40.6                |
> | + PTClf          | **88.6** (**+1.2**) | **48.0** (**+0.8**) | **41.2** (**+0.6**) |
> | ViT-H/14         | 88.6                | 47.9                | 44.3                |
> | + PTClf          | **89.5** (**+0.9**) | **48.6** (**+0.7**) | **45.1** (**+0.8**) |
> | Swin-B           | 83.5                | 48.1                | 36.7                |
> | + PTClf          | **84.9** (**+1.4**) | **49.2** (**+1.1**) | **37.7** (**+1.0**) |
> | CLIP-ViT-B/16    | 77.2                | 51.0                | 42.3                |
> | + PTClf          | **78.2** (**+1.0**) | **51.6** (**+0.6**) | **43.4** (**+1.1**) |
> | DINOv2-B         | 86.1                | 47.2                | 51.4                |
> | + PTClf          | **87.2** (**+1.1**) | **48.0** (**+0.8**) | **52.1** (**+0.7**) |
>
> As shown, PTClf consistently improves performance across various architectures, indicating that its superior generalizability. Regarding self-supervised models, the DINOv2-B model used here includes an ImageNet linear-probe classifier [1], so our method can be applied directly. We acknowledge that purely self-supervised models pre-trained without classifiers pose a distinct challenge for PTClf. *In practice, however, many popular self-supervised models release ImageNet linear-probe checkpoints that contain classifiers* \[1-4], enabling straightforward application of our approach. Nevertheless, we agree that extending PTClf to self-supervised models without any classifier (even a linear probe) remains an important direction for our future work.
>
> [1] https://huggingface.co/facebook/dinov2-base-imagenet1k-1-layer.
>
> [2] https://huggingface.co/timm/hiera_base_plus_224.mae_in1k_ft_in1k.
>
> [3] https://huggingface.co/BobMcDear/vit_base_patch16_mae_in1k_224.
>
> [4] https://huggingface.co/lightly-ai/simclrv1-imagenet1k-resnet50-1x.
>
>
> &nbsp;
>
> > **[W3].** Results in Table ii lag behind those reported in [1], which used a ResNet-50 backbone five years ago (e.g., Stanford Cars vanilla fine-tuning reaches 87.20%, whereas the highest in Table iii is 59.6%). It is unclear whether evaluation protocols differ in the fine-grained fine-tuning setup.
>
> **We clarify that all FGVC datasets in our expeirments (including Stanford Cars) were adapted to long-tailed versions instead of using their original balanced form, as noted in Sec. 5.1.** In the long-tailed version of Stanford Cars, 20 classes contain only a single sample each, causing severe tail-class scarcity and naturally leading to lower overall accuracy even with long-tailed strategies. To directly address your concern, we also conduct experiments on the full, standard Stanford Cars dataset. The results are presented in the table below, where '*' denotes applying data augmentation.
>
> |                               | Pre-Trained Model | Stanford Cars |
> | ----------------------------- | ----------------- | ------------- |
> | Baseline (Adapter+)           | ViT-B/16          | 83.8          |
> | + PTClf                       | ViT-B/16          | 84.8          |
> | SSF [2]                       | ViT-B/16          | 82.6          |
> | VPT [2]                       | ViT-B/16          | 83.6          |
> | Baseline (Adapter+)*          | ViT-B/16          | 89.0          |
> | + PTClf*                      | ViT-B/16          | **89.8**      |
> | Vanilla Full Fine-Tuning* [1] | ResNet-50         | 87.2          |
> | Co-Tuning (Adapter+)*         | ViT-B/16          | 89.4          |
>
> As shown, our results on the full Stanford Cars dataset are comparable to those reported in [1], as well as to other PEFT-based methods [2]. **Remarkably, this pronounced gap between the full and long-tailed settings further underscores the importance of addressing real-world long-tailed scenarios, which can significantly compromise model performance, even when using strong foundation models with advanced long-tailed techniques.**
>
> [1] You, Kaichao, et al. Co-tuning for transfer learning. NeurIPS'20.
>
> [2] Ye, Peng, et al. Partial fine-tuning: A successor to full fine-tuning for vision transformers. ArXiv'23.

---

> ### Author Response · Authors · 2025-11-23
> **Rebuttal by Authors [3/3]**
>
> > **[Q1].** The authors claim to focus on PEFT, long-tailed, visual tasks, but the methodology appears general. Can PTClf be extended to other domains, such as LLM fine-tuning?
>
> **Indeed, our proposal is general for various classification tasks, including LLM-based classification.** Following [1], we further conduct experiments on the CoNNL2003 named entity recognition task (i.e., a token-level classification problem in NLP) by fine-tuning BERT. The results are presented in the table below:
>
> |           | Baseline | Co-Tuning   | PTClf       | Co-Tuning + PTClf   |
> | --------- | -------- | ----------- | ----------- | ------------------- |
> | CoNLL2003 | 90.8     | 91.3 (+0.5) | 91.9 (+1.1) | **92.2** (**+1.4**) |
>
> As shown, PTClf still delivers superior performance in the context of LLM fine-tuning.
>
> [1] You, Kaichao, et al. Co-tuning for transfer learning. NeurIPS'20.
>
>
>
> &nbsp;
>
> > **[Q2].** Can PTClf be applied to tasks beyond classification? The long-tailed problem is even more challenging in object detection.
>
> Thank you for your valuable question. To investigate this, we conduct preliminary experiments on object detection using PASCAL VOC, based on a COCO-trained Faster R-CNN with a VGG-16 backbone. The results are presented in the table below:
>
> |            | Baseline | PTClf       |
> | ---------- | -------- | ----------- |
> | PASCAL VOC | **77.5** | 77.4 (-0.1) |
>
> As shown, directly applying PTClf to object detection exhibits negligible changes. We attribute this to a fundamental difference between classification and detection: in image classification, the classifier operates on global features, whereas in object detection, the classifier operates on local features corresponding to each region of interest. This finer feature granularity may naturally limit the transferability of the learned class anchors. Nevertheless, we agree that long-tailed object detection is an important and challenging direction. Developing a detection-tailored extension of PTClf is an important direction we intend to pursue.
>
>
> &nbsp;
>
>
> > **[Q3].** The assumption that the pre-trained model covers more classes usually does not hold (line 256). In your fine-grained benchmarks—e.g., Stanford Dogs, a subset of ImageNet with finer labels—the downstream task is more fine-grained than the pre-trained task. How do you ensure that the label mapping works well in this setup?
>
> **The label mapping does not require the upstream label space to be identical or a superset of the downstream one; instead, it automatically selects the most semantically related upstream class for each downstream class.** To illustrate this, we visualize the mapping dynamics on four datasets with varying degrees of label-space overlap:
>
> - *Stanford Dogs (complete overlap)* is a subset of ImageNet, where each downstream class corresponds exactly to an ImageNet class rather than to a finer label. As shown in the visualization (https://drive.google.com/file/d/1f4TmDaIkQ8jfB1n_ePO5eiRzRgr3_YLx), the mapping easily matches the exact corresponding ImageNet label at the initial stage.
> - *CIFAR100 (partially overlap)* contains natural images and shares only part of its label space within ImageNet. As shown in the visualization (https://drive.google.com/file/d/1dHwdLmDJaKVigm-1NReW4kzud7kPue7u), the mapping effectively aligns each downstream class to a semantically related ImageNet class.
> - *DTD (largely disjoint)* comprises texture-based categories with no direct counterparts in ImageNet. As shown in the visualization (https://drive.google.com/file/d/1GdJdQ5T_eD_28UT-Os6oRZMmgNYOK9oU), the mapping still converges to ImageNet classes that exhibit similar visual or semantic attributes.
> - *EuroSAT (completely disjoint)* consists of satellite imagery, which is even more distant from ImageNet. As shown in the visualization (https://drive.google.com/file/d/1zHYaog_QDi5eUAS1tspEtwJJyD_p3qb5), the mapping still aligns each land‑use class with ImageNet classes that exhibit analogous global structure or layout.
>
> Across these four datasets, ranging from fully overlapping (Stanford Dogs) to fully disjoint (EuroSAT), the learned mapping consistently pairs downstream labels with semantically related upstream labels. This suggests that even when the downstream task lies in a different domain, the proposed label‑mapping strategy can still capture meaningful semantic relationships and provide appropriate class anchors for effective transfer.

---

### Official Review · Reviewer_X5HW · 2025-11-01

**Soundness:** 3
**Presentation:** 3
**Contribution:** 3
**Rating:** 6
**Confidence:** 3

**Summary:**

The authors propose PTClf (Pre-Trained Classifier helps), a novel fine-tuning paradigm that leverages pre-trained classifiers to assist re-trained classifiers in learning tail classes. It aligns downstream classes to upstream classes via label mapping, and guide the re-trained classifier to learn from the mapped pre-trained classifier through initialization and regularization, thereby transferring knowledge from related upstream classes to the data-scarce tail classes.
Experimental results across 8 datasets show consistent and strong improvements, particularly for tail classes.

**Strengths:**

- The motivation of reusing pre-trained classifiers for long-tailed learning is interesting and inspiring.
- The introduction of CDI and GI metrics for channel-wise classifier analysis reveals specific failure modes of re-trained classifiers (sparse discriminative channels, reliance on general features).
- The method demonstrates consistent improvements across various benchmarks, especially on tail classes, along with extensive ablation studies.
- The paper is well delivered and very easy to follow.

**Weaknesses:**

- The evaluation focuses heavily on ViT-B/16 with limited architectural diversity.
- The simple frequency-based label mapping may be suboptimal compared to more sophisticated alignment methods (e.g., embedding similarity, semantic matching). The approach also lacks analysis of failure cases or guidance on when the method works versus when it doesn't.

**Questions:**

- How could the method reuse more general pre-trained zero-shot classifiers like CLIP?
- Can the authors discuss on relation to papers like LIFT+ (Lightweight Fine-Tuning for Long-Tail Learning)?

---

> ### Author Response · Authors · 2025-11-23
> **Rebuttal by Authors [1/3]**
>
> Thank you for your positive recommendation and constructive feedback. We address your concerns and questions below:
>
> > **[W1].** The evaluation focuses heavily on ViT-B/16 with limited architectural diversity.
>
> **PTClf exhibits superior generalizability across diverse architectures.** We conduct additional experiments using (i) ViT-based architectures (ViT-L/16 and ViT-H/14), (ii) Swin-based architecture (Swin-B), (iii) CLIP-based architecture (CLIP-ViT-B/16), and (iv) DINO-based architecture (DINOv2-B equipped with a pre-trained classifier [1]). The results are presented in the following table:
>
> |                  | C100-IR200          | Places-LT           | FGVCAircraft-LT     |
> | ---------------- | ------------------- | ------------------- | ------------------- |
> | ViT-B/16 (paper) | 85.2                | 45.9                | 30.7                |
> | + PTClf          | **88.1** (**+2.9**) | **47.9** (**+2.0**) | **32.7** (**+2.0**) |
> | ViT-L/16         | 87.4                | 47.2                | 40.6                |
> | + PTClf          | **88.6** (**+1.2**) | **48.0** (**+0.8**) | **41.2** (**+0.6**) |
> | ViT-H/14         | 88.6                | 47.9                | 44.3                |
> | + PTClf          | **89.5** (**+0.9**) | **48.6** (**+0.7**) | **45.1** (**+0.8**) |
> | Swin-B           | 83.5                | 48.1                | 36.7                |
> | + PTClf          | **84.9** (**+1.4**) | **49.2** (**+1.1**) | **37.7** (**+1.0**) |
> | CLIP-ViT-B/16    | 77.2                | 51.0                | 42.3                |
> | + PTClf          | **78.2** (**+1.0**) | **51.6** (**+0.6**) | **43.4** (**+1.1**) |
> | DINOv2-B         | 86.1                | 47.2                | 51.4                |
> | + PTClf          | **87.2** (**+1.1**) | **48.0** (**+0.8**) | **52.1** (**+0.7**) |
>
> As shown, PTClf consistently improves performance across different architectures.
>
> [1] https://huggingface.co/facebook/dinov2-base-imagenet1k-1-layer.

---

> ### Author Response · Authors · 2025-11-23
> **Rebuttal by Authors [2/3]**
>
> > **[W2].** The simple frequency-based label mapping may be suboptimal compared to more sophisticated alignment methods (e.g., embedding similarity, semantic matching).
>
> **Indeed, naive frequency-based label mapping is not the most effective design choice.** Following the reviewer's suggestion, we further introduce three alternative mapping strategies:
>
> - *Semantic Matching.* We compute label semantic similarity between downstream and upstream classes by LLMs to establish alignment.
> - *Anchor Similarity*. Since upstream samples or embeddings are not available for downstream fine-tuning, we instead aligns labels based on the similarity between class anchors of the re-trained and pre-trained classifiers.
> - *Probability-based Mixed Mapping*. Rather than assigning only a single pre-trained class anchor to each downstream class, we construct new mixed anchors for the re-trained classifier by aggregating pre-trained anchors in a probability-weighted manner. Specifically, we generate them by: ${\bf{w}} ^ {\prime} _ {k ^ \text{r}} = \sum _ {k ^ \text{p} \in [K ^ \text{p}]} p _ {k ^ \text{p}(k ^ \text{r})} {\bf w} _ {k ^ \text{p}}$, where $p _ {k ^ {\text{p}} (k ^ {\text{r}})}$ denotes the prediction probabilities (i.e., normalized frequency).
>
> |                                 | C100-IR200      | Places-LT           | FGVCAircraft-LT     |
> | ------------------------------- | ------------------- | ------------------- | ------------------- |
> | Baseline                        | 85.2                | 45.9                | 30.7                |
> | Frequency-based Mapping (paper) | 88.1 (+2.9)         | 47.9 (+2.0)         | 32.7 (+2.0)         |
> | Semantic Matching               | 87.6 (+2.4)         | 48.4 (+2.5)         | **32.9** (**+2.2**) |
> | Anchor Similarity               | 82.6 (-2.6)         | 44.9 (-1.0)         | 30.2 (-0.5)         |
> | Probability-based Mixed Mapping | **88.7** (**+3.5**) | **48.5** (**+2.6**) | 31.8 (+1.1)         |
>
> As shown in the table, both semantic matching and probability-based mapping slightly outperform the frequency-based strategy. **Nevertheless, our goal is to design a simple yet effective approach, rather than rely on more sophisticated alignment strategies.** For this reason, we keep adopting the frequency-based mapping as our default design choice.
>
> &nbsp;
>
> > **[W2].** The approach also lacks analysis of failure cases or guidance on when the method works versus when it doesn't.
>
> **Our method exhibits failure cases on extremely OOD data like medical imagery.** We conduct additional experiments on three highly OOD datasets: ISIC-LT (skin lesions), EuroSAT-LT (satellite imagery), and RareSpecies-LT (rare biological species). These datasets are completely disjoint from the upstream ImageNet and thus represent extreme OOD settings.
>
> |          | ISIC-LT     | EuroSAT-LT          | RareSpecies-LT      |
> | -------- | ----------- | ------------------- | ------------------- |
> | Baseline | **69.8**    | 89.5                | 53.6                |
> | + PTClf  | 69.7 (-0.1) | **89.9** (**+0.4**) | **54.4** (**+0.8**) |
>
> As shown in the table, PTClf becomes less effective in these settings, even exhibiting a slight performance drop on ISIC-LT. We attribute this to the sharp mismatch between the downstream label space and ImageNet's natural-image labels. When label semantics are largely orthogonal, ImageNet-derived class anchors offer limited benefit for the downstream classes, thereby constrainting their effectiveness. Nevertheless, owing to large-scale pre-training, transferring related knowledge from upstream classifiers remains beneficial in most cases, including on moderately OOD datasets like SVHN and DTD, and even for parts of highly OOD datasets like RareSpecies.

---

> ### Author Response · Authors · 2025-11-24
> **Rebuttal by Authors [3/3]**
>
> > **[Q1].** How could the method reuse more general pre-trained zero-shot classifiers like CLIP?
>
> **PTClf can indeed be extended to reuse more general zero-shot models like CLIP.** Although CLIP does not contain an explicit pre-trained classifier, its zero-shot predictions are obtained from the text embeddings of class names, which play the same role as upstream class anchors. Consequently, we reuse the linguistic knowledge within CLIP by generating text embeddings for the IN-21k class names as pre-trained anchors. These anchors are then used to regularize the fine-tuning of the text encoder in the same manner as the pre-trained classifier in PTClf. The results are presented in the table below:
>
> |         | C100-IR200          | Places-LT           | FGVCAircraft-LT     |
> | ------- | ------------------- | ------------------- | ------------------- |
> | CLIP    | 77.2                | 51.0                | 42.3                |
> | + PTClf | **78.2** (**+1.0**) | **51.6** (**+0.6**) | **43.4** (**+1.1**) |
>
> As shown, PTClf also brings consistent gains when reusing the knowledge within the general zero-shot model like CLIP.
>
> *For simplicity, we directly use the IN-21k class names without additional curation.
>
> &nbsp;
>
>
> > **[Q2].** Can the authors discuss on relation to papers like LIFT+ (Lightweight Fine-Tuning for Long-Tail Learning)?
>
> LPT [1] and LIFT+ [2] are the most relevant and advanced works compared to ours. Specifically, LPT is a ViT-based method that integrates various long-tailed techniques (e.g., cosine classifier, mixup, gcl [3], and re-weighting [4]) with prompt tuning, and LIFT+ is a CLIP-based method that combines long-tailed strategies (e.g., logit adjustment [5]) with different PEFT techniques. **Both methods primarily focus on incorporating long-tailed strategies into fine-tuning** to mitigate imbalance while leveraging the pre-trained model. In contrast, **PTClf centers on the fine-tuning paradigm itself**, aiming to better utilize the knowledge within the pre-trained model for long-tailed learning. Following the extension strategy described in Q1, we also evaluate PTClf combined with the CLIP-based method LIFT+. The results are provided in the table below:
>
> |           | C100-IR200          | Places-LT           | FGVCAircraft-LT     |
> | --------- | ------------------- | ------------------- | ------------------- |
> | LIFT+ [2] | 78.6                | 51.5                | 43.1                |
> | + PTClf   | **79.4** (**+0.8**) | **52.1** (**+0.6**) | **44.1** (**+1.0**) |
>
> As shown, PTClf also brings consistent improvements when applied to LIFT+.
>
> [1] Dong, Bowen, et al. Lpt: Long-tailed prompt tuning for image classification. ICLR'23.
>
> [2] Shi, Jiang-Xin, Tong Wei, and Yu-Feng Li. LIFT+: Lightweight Fine-Tuning for Long-Tail Learning. ArXiv'25.
>
> [3] Li, Mengke, et al. Adjusting logit in Gaussian form for long-tailed visual recognition. TPAMI'24.
>
> [4] Cao, Kaidi, et al. Learning imbalanced datasets with label-distribution-aware margin loss. NeurIPS'19.
>
> [5] Menon, Aditya Krishna, et al. Long-tail learning via logit adjustment. ICLR'21.

---

### Author Response · Authors · 2025-12-03
**General Response**

We sincerely appreciate the reviewers for their thoughtful and constructive feedback. We are encouraged by the positive recognition of our contributions, which can be summarized as follows:

1. Our analysis of classifier bias in long-tailed learning is solid and interesting (Reviewers `X5HW`, `CDwk`, `EgUh`)
2. The proposed PTClf approach flexible and easy to follow (Reviewers `X5HW`, `6hC2`, `EgUh`).
3. Our method demonstrates superior empirical performance across multiple benchmarks (Reviewers `X5HW`, `CDwk`, `6hC2`, `EgUh`).
4. Our experimental study on the pre-trained classifier is thorough carefully executed (Reviewers `X5HW`, `6hC2`, `EgUh`).

In our revision, we have carefully addressed the reviewers' concerns. We specifically summarize three main concerns and our corresponding responses as follows:

| Reviewers' Concerns | Authors' Responses |
| --- | ----- |
| The paper's novelty is limited, since previous study (i.e., Co-Tuning) has already explored leveraging the pre-trained classifier. (Reviewer `CDwk`) | We highlight the novelty of our work from two perspectives. **(1) Our work focuses on the *classifier*, in clear contrast to Co-Tuning.** Co-Tuning uses the pre-trained classifier to co-supervise the backbone in a multi-task learning manner, aiming to improve representation learning, yet its classifier is still *re-trained from scratch without guidance*. Instead, our method uses the pre-trained classifier to directly guide the re-trained classifier via class-to-class knowledge transfer, aiming to mitigate classifier bias in long-tailed learning and improve classifier learning. **(2) Our contribution goes beyond methodology to a deeper analysis of long-tailed classifiers.** Classifier bias has long been recognized as a core challenge in long-tailed learning, but prior analyses have mainly focused on norm disparities and **attributed it to class imbalance**. By introducing channel-level metrics, we go one step deeper to reveal weakened discriminative ability and semantic awareness as key underlying factors, and further show that both **stem mainly from sample scarcity in tail classes, rather than imbalance itself (Appendix B)**. We believe these findings provide meaningful insights for the long-tailed learning community beyond the method itself. |
| The paper lacks evaluations on a broader range of architectures, particularly CLIP and self-supervised models, which undermines it practical applicability. (Reviewers `X5HW`, `CDwk`, `6hC2`, `EgUh`) | We have added additional experiments on diverse architectures, including: **(1) ViT & Swin.** We evaluate PTClf on larger-scale ViTs and Swin Transformer. **(2) CLIP.** While CLIP does not provide an explicit pre-trained classifier, PTClf can still be naturally extended to this setting. Specifically, the core idea behind PTClf is to leverage upstream knowledge for data-scarce classification, which can be applied to CLIP by exploiting the linguistic priors within the text encoder. **(3) Self-Supervised Models.** While purely self-supervised models trained without classifier pose a distinct challenge, PTClf can be applied to linear-probe self-supervised checkpoints, thereby broadening its practical applicability to self-supervised settings.  As shown in **Appendix E.1**, PTClf consistently improves performance across these different architectures. |
| PTClf's gains may stem from label overlap or better classifier properties, rather than from knowledge transfer. (Reviewers `CDwk`, `6hC2`, `EgUh`) | We have added further analyses in **Appendix C** and summarize the key findings as follows. **(1) PTClf benefits from both overlapping and non-overlapping classes (Insight 7).** We quantify the semantic relatedness between upstream and downstream labels by WordNet path similarity and divide them into Overlap, Similar, and Distant groups. Across all three groups, PTClf consistently improves performance for both overlapping and non-overlapping classes. **(2) PTClf's gains are not simply due to inherently better classifier properties (Insight 8).** We replace our label mapping with a random mapping, which preserves the potential advantages of better classifier properties while removing label correlations. The resulting performance drop confirms that PTClf's gains cannot be simply attributed to better classifier properties. **(3) Knowledge transfer from semantically related upstream classes is the main source of PTClf's gains (Insight 9).** We visualize the mapping dynamics across datasets from highly OOD to ID. Two patterns emerge: (i) PTClf's gains clearly scale with the knowledge correlation between upstream and downstream labels; and (ii) even for OOD datasets, the learned mappings tend to align downstream labels with semantically related upstream classes. Togerther, these observations support that PTClf's benefits largely arise from effective knowledge transfer via semantically related upstream classes. |

---

### Meta-Review · Area_Chair_5Ue7 · 2025-12-28

**Summary:**

This submission received **mixed but overall below-threshold initial scores**, with one reviewer recommending marginal acceptance (Reviewer X5HW) and all other reviewers assigning marginal reject scores (Reviewers CDwk, 6hC2, EgUh). While reviewers generally agree that the paper is well written, empirically thorough, and studies an important problem (long-tailed learning), the majority raised consistent concerns regarding limited novelty, dependence on pretraining proximity, and insufficiently established generality.

After considering the rebuttal and additional experiments, my overall judgment is that the core concerns remain unresolved. The added results strengthen the empirical section but do not fundamentally change the assessment of the paper’s contribution level or scope. Therefore, my recommendation is **Reject**.

**Reviewer Concerns:**

**Concerns partially addressed by the rebuttal**

* **Architectural diversity and applicability beyond ViT-B/16**:
  The authors added experiments on ViT-L/16, ViT-H/14, Swin-B, CLIP-ViT-B/16, and DINOv2-B, showing consistent gains across architectures
  *(Reviewers X5HW, CDwk, EgUh)*.

* **Overlap vs. knowledge transfer**:
  The rebuttal includes random label mapping experiments and WordNet-based semantic grouping, suggesting that PTClf’s gains correlate with semantic relatedness rather than purely with classifier initialization or overlap
  *(Reviewers 6hC2, EgUh, CDwk)*.

* **Clarification of evaluation protocols and FGVC discrepancies**:
  The authors clarified that several datasets are long-tailed variants and added results on the full Stanford Cars dataset to contextualize absolute accuracy gaps
  *(Reviewer CDwk)*.


**Concerns that remain outstanding**

* **Dependence on upstream–downstream similarity**:
  Reviewers consistently noted that PTClf’s effectiveness appears strongly tied to semantic or visual proximity to ImageNet-style pretraining. The newly added extreme-OOD results (e.g., ISIC-LT) show very limited gains
  *(Reviewers 6hC2, EgUh)*.

* **Evidence for knowledge transfer remains indirect**:
  Although the random-mapping control is helpful, it remain unconvinced that the paper fully disentangles semantic knowledge transfer from other effects such as regularization, calibration, or optimization ease due to initialization.  *(Reviewers EgUh, 6hC2)*.

* **Limited scope beyond classification**:
  Preliminary object detection experiments show negligible improvement, suggesting that the approach does not readily extend beyond global classification heads. This limits the broader impact, especially since long-tailed problems are prominent in detection and instance-level tasks
  *(Reviewer CDwk)*.

**Reviewer Scores:**

* **Reviewer X5HW (initial score: 6)** → **Likely unchanged**
  Positive on analysis, clarity, and empirical consistency; rebuttal reinforces but does not shift the overall committee balance.

* **Reviewer CDwk (initial score: 4)** → **Likely unchanged**
  Novelty concerns persist despite added experiments and clarifications. And the application of the proposed method is limited to just classification.

* **Reviewer 6hC2 (initial score: 4)** → **Likely unchanged**
  Core concerns about dependence on semantic overlap and limited OOD generalization remain.

* **Reviewer EgUh (initial score: 4)** → **Likely unchanged**
  Practicality and strength of the knowledge-transfer claim remain insufficiently demonstrated.

---

### Decision · Program_Chairs · 2026-01-26

Reject